# FISTAPRUNER: LAYER-WISE POST-TRAINING PRUNING FOR LARGE LANGUAGE MODELS

## ABSTRACT

Pruning is a critical strategy for compressing trained large language models (LLMs), aiming at substantial memory conservation and computational acceleration without compromising performance. However, existing pruning methods typically necessitate inefficient retraining for billion-scale LLMs or rely on heuristically designed metrics to determine pruning masks, leading to performance degradation. This paper presents, for the first time, a LASSO-like convex optimization model crafted to induce sparsity in LLMs. By leveraging the FISTA, we introduce FISTAPruner, a novel method that includes a cumulative error elimination mechanism within decoder layers and supports parallel pruning for unstructured pruning. Additionally, we extend this method to 2:4 semi-structured pruning. We comprehensively evaluate FISTAPruner on models such as OPT and LLaMA variants with 125M to 70B parameters under unstructured and 2:4 semi-structured sparsity, showcasing superior performance over existing methods across various language benchmarks. Notably, it can remove 50% of the model parameters for LLaMA-3-70B while retaining 98.6% and 95.6% of the zero-shot task performance under these two sparsity patterns, respectively.

## 1 INTRODUCTION

In recent years, large language models (LLMs) have revolutionized natural language processing fields, achieving impressive results in tasks such as machine translation, sentiment analysis, question answering, and text generation (Lyu et al., 2023; Yao et al., 2023; Zhang et al., 2023a;b; Wang et al., 2023; Arefeen et al., 2024; Li et al., 2024). Advanced LLMs such as OpenAI's GPT-4 (OpenAI, 2023), Meta's LLaMA-3 (Meta AI, 2023), and Google's Gemini (Gemini Team et al., 2023) excel in generating coherent text with extensive parameters. However, the growth in model sizes outpaces hardware improvements, posing significant deployment and inference challenges (Steiner et al., 2023). For example, operating OPT-175B (Zhang et al., 2022) requires over 320GB of memory and at least five 80GB A100 GPUs for loading its parameters in FP16 precision. This challenge becomes more pronounced in environments with limited resources, such as mobile devices, edge computing systems, and real-time applications. Consequently, there has been considerable interest in compressing LLMs to enhance their efficiency and practicality for deployment across various applications.

Pruning is a key method for compressing LLMs, aiming to eliminate redundant weights to reduce model size and computational demands while striving to maintain performance. Methods such as those in (Huang et al., 2020; Ma et al., 2023; Zhang et al., 2023c) require a retraining phase post-pruning, which is inefficient for billion-scale LLMs. PERP (Zimmer et al., 2023) introduces an efficient retraining approach after pruning to recover the performance of pruned model. Recent developments, including SparseGPT (Frantar & Alistarh, 2023) and Wanda (Sun et al., 2023), employ post-training pruning techniques for LLMs without retraining. These methods, however, rely on the heuristic-based optimal brain surgeon (OBS) framework (Hassibi & Stork, 1992) or utilize heuristic-based pruning metrics to determine pruning masks, potentially compromising performance. DSnoT (Zhang et al., 2023d) introduces a training-free fine-tuning approach that updates the results of other pruning methods, such as SparseGPT and Wanda, which also depend on heuristic-based adjustment metrics.

In this work, we first introduce a LASSO-like convex optimization model for layer-wise post-training unstructured pruning of LLMs. Figure 1 provides an overview of our method, which is applied to each linear operator. We employ the Frobenius norm of the difference between the outputs obtained

Figure 1: Overview of the proposed FISTAPruner. Given a weight matrix $\boldsymbol{W}$ and its corresponding input feature activation $\boldsymbol{X}$, we employ the proposed convex optimization model, utilizing FISTA, to derive the pruned weights.

from the dense and pruned weights to quantify the output error. Additionally, we integrate an $\ell_1$-norm regularization term, the optimal convex approximation of the $\ell_0$-norm (Candès et al., 2006), into each row of weights to promote sparsity. The solutions of the proposed optimization model demonstrate a balanced trade-off between output error and sparsity, governed by our proposed adaptive tuning method that meticulously adjusts the hyperparameter $\lambda$. To solve this optimization problem efficiently, we utilize the Fast Iterative Shrinkage-Thresholding Algorithm (FISTA) (Beck & Teboulle, 2009), which ensures a convergence rate of $O(1/k^2)$. Following this, we name our proposed method FISTAPruner. We further extend it to accommodate 2:4 semi-structured pruning by incorporating a hard thresholding step following FISTA's convergence, thus achieving the desired sparsity structures.

In addition, our approach effectively mitigates the cumulative error within decoder layers resulting from pruning by incorporating an intra-layer error correction mechanism. Due to discrepancies between the outputs of dense and pruned weights, errors can accumulate, as the output from one pruned weight becomes the input for the next operator. FISTAPruner addresses this by sequentially pruning the weights of each linear operator within a decoder layer, using the output from the pruned weights of one operator as the input for the next, thus minimizing output discrepancies. Additionally, FISTAPruner treats each decoder layer as an independent unit for pruning, allowing for the simultaneous pruning of multiple decoder layers and significantly increasing efficiency.

We empirically evaluate FISTAPruner on the widely adopted OPT (Zhang et al., 2022), LLaMA (Touvron et al., 2023a), and LLaMA-2 (Touvron et al., 2023b) model families, as well as the latest LLaMA-3 (Touvron et al., 2023a) models. FISTAPruner's layer-by-layer pruning implementation allows for the pruning of these LLMs ranging from 125M to 70B parameters on a single NVIDIA A100 GPU with 40GB of memory. Our results confirm that FISTAPruner can efficiently create sparse networks from pretrained LLMs without retraining. Moreover, our approach exceeds the performance of state-of-the-art methods such as SparseGPT, Wanda, DSnoT, and PERP across various language benchmarks. We also perform a series of ablation studies to validate our methods. We believe our work sets a new direction and baseline for future research in this area and encourages further exploration into understanding sparsity in LLMs with the tools of convex optimization.

## 2 BACKGROUND AND RELATED WORK

**Pruning of LLMs.** Pruning is a widely used strategy to compress LLMs by generating sparse weight matrices under unstructured, semi-structured, and structured sparsity based on calibration data. Unstructured sparsity of rate $s\%$, eliminates $s\%$ of the entries in a weight matrix. Semi-structured sparsity with proportion $n : m$ maintains a fixed overall sparsity level $n/m$, and allows at most $n$ non-zero entries in every group of $m$ consecutive entries. Pruning weights into semi-structured sparsity, especially with proportion 2:4, could yield up to $2\times$ inference speedup using NVIDIA GPUs with the Ampere architecture (Mishra et al., 2021) and hence is of particular interest. Structured sparsity, which zeroes entire rows or columns, offers significant computational and memory benefits but can lead to greater performance losses.

**Pruning with Retraining.** Traditional pruning pipelines often include a retraining step to offset performance losses (Huang et al., 2020; Ma et al., 2023; Zhang et al., 2023c). However, the sheer scale of LLMs makes this additional retraining costly in terms of both time and computational resources. Dinh et al. (2020); Holmes et al. (2021); Xie et al. (2023) integrate retraining directly into the pruning process by targeting the minimization of the highly non-convex loss function related

to the calibration dataset, using the alternating direction method of multipliers (ADMM) to derive pruned weights. Nonetheless, this approach imposes significant computational demands and the use of ADMM in non-convex optimization often results in unstable performance (He & Yuan, 2012).

**Pruning without Retraining.** Pruning without retraining offers a straightforward alternative, eliminating the need for post-pruning retraining. These methods prune LLMs in a single step, simplifying implementation and reducing both time and computational demands. Consequently, various methods have been developed under different sparsity frameworks. For structured pruning, SliceGPT (Ashkboos et al., 2024) and Eigenpruning (Vergara-Browne et al., 2024) utilize singular value decompositions to prune singular values of weight matrices and reduce model dimensions. ZipLM (Kurtić et al., 2024) adopts an OBS-based approach for structured pruning and updates remaining weights to maintain performance. Our proposed FISTAPruner focuses on unstructured and semi-structured pruning, and thus is orthogonal to these structured pruning methods, enabling further model compression. For unstructured and semi-structured pruning, SparseGPT (Frantar & Alistarh, 2023) and ISC (Shao et al., 2024) leverage the OBS framework to calculate saliency for each entry using the inverse Hessian of the loss metric, based on which pruning masks are generated and weights updated. Wanda (Sun et al., 2023) implements a heuristic approach, removing weights based on the product of their magnitudes and activations without compensation. DSnoT (Zhang et al., 2023d) updates the results of other pruning methods, such as SparseGPT and Wanda, which also relies on heuristic-based adjustment metrics. Boža (2024) employs ADMM to optimize weight updates under iteratively refined pruning masks chosen through heuristic methods based on Wanda. These strategies adopt a layer-wise pruning strategy, where errors between the pruned output and the original output of each operator accumulates. Moreover, due to their heuristic nature, the performances of the pruned models are unstable and compromised.

**Error Corrections.** Error correction techniques are increasingly used to mitigate error accumulations from layer-wise pruning by minimizing reconstruction errors between the pruned network and the original one (Park et al., 2024; El Halabi et al., 2022). However, their implementations and applications to pruning LLMs vary widely. Prominent methods like SparseGPT (Frantar & Alistarh, 2023) focus on pruning without explicit error correction, while approaches like K-prune (Park et al., 2024) minimize global reconstruction error, facing scalability challenges as globally correcting pruning errors will require global sequential pruning. Our work introduces intra-layer error corrections for better accuracy and computational efficiency. By focusing on intra-layer adjustments, our method provides a scalable and effective solution for pruning LLMs.

## 3 METHODOLOGY

In this section, we introduce our post-training pruning method, FISTAPruner, which comprises three main components. First, we address the error accumulation issue in layer-wise pruning with an intra-layer error correction mechanism and develop a novel convex optimization model tailored for this purpose. We then detail the process for unstructured pruning using FISTA and adapt the framework for $n : m$ semi-structured pruning. Finally, we present an adaptive method that finely tunes the hyperparameter $\lambda$ in our model to minimize the output discrepancies between dense and pruned operators while achieving the desired sparsity level.

### 3.1 POST-TRAINING PRUNING MODEL WITH INTRA-LAYER ERROR CORRECTIONS

Post-training compression is typically achieved by decomposing the full-model compression problem into layer-wise subproblems (Frantar & Alistarh, 2023). For instance, a typical Transformer decoder layer (Vaswani et al., 2017) comprises six crucial linear operators: $\boldsymbol{W}_Q$, $\boldsymbol{W}_K$, $\boldsymbol{W}_V$, $\boldsymbol{W}_O$, $\boldsymbol{W}_{fc_1}$, and $\boldsymbol{W}_{fc_2}$. We leverage an intra-layer error correction mechanism that sequentially prunes the weights while explicitly accounting for the cumulative error introduced at each step. Consider a dense weight matrix $\boldsymbol{W} \in \mathbb{R}^{m \times n}$ and the corresponding input activation $\boldsymbol{X} \in \mathbb{R}^{n \times p}$. The output is $\boldsymbol{Z} = \boldsymbol{W}\boldsymbol{X}$. Our goal is to find the pruned weights $\boldsymbol{W}^*$ that minimize the discrepancy between the outputs of the dense and pruned models:

$$\min_{\boldsymbol{W}^*} \|\boldsymbol{W}^*\boldsymbol{X}^* - \boldsymbol{W}\boldsymbol{X}\|_F^2 \quad \text{subject to} \quad \boldsymbol{W}^* \in \mathcal{S}, \tag{1}$$

where $\|\cdot\|_F$ denotes the Frobenius norm, and $\mathcal{S}$ defines the permissible sparsity patterns. The input activation $\boldsymbol{X}^*$ are defined based on the position of the operator within the layer. Specifically, if the

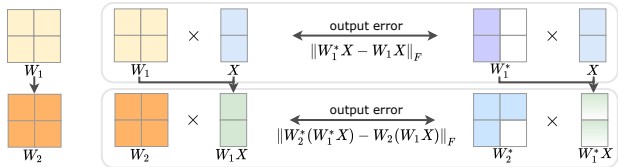

Figure 2: Illustration of the proposed intra-layer error correction mechanism. $W_1$ and $W_2$ represent the weights of two sequential layers within the network architecture.

operator is at the top of the layer, then $X^* = X$. Conversely, if the operator follows previously pruned operators, $X^*$ is set to $Z^*_{\text{prev}}$, where $Z^*_{\text{prev}}$ is the pruned output from the preceding operator. As illustrated in Figure 2, consider two sequential operators with weights $W_1$ and $W_2$. When pruning $W_1$ to obtain its pruned counterpart $W_1^*$, Equation 1 quantifies the output error between $W_1 X$ and $W_1^* X$, where the input $X^*$ remains the same as $X$ since this operator is at the top of the layer. However, for the second operator $W_2^*$, the corresponding input becomes $W_1^* X$ instead of $W_1 X$ due to the pruning applied to $W_1$. Consequently, the deviation between the outputs of $W_2$ and $W_2^*$ is computed by comparing $W_2(W_1 X)$ and $W_2^*(W_1^* X)$. This approach ensures that cumulative error is appropriately considered, as each pruning step accounts for both the changes in the weights and the modified input activations resulting from previous pruning. Note that we use intra-layer error corrections within each decoder layer, enabling parallel pruning and improved performance (see Section 4.4 for details).

Unstructured pruning essentially transforms dense weight matrices into sparse structures. The $\ell_0$-norm, which directly counts the number of non-zero entries in a vector, is the most straightforward measure of unstructured sparsity. Despite the intuitive appeal of the $\ell_0$-norm, it induces non-convex and NP-hard optimization challenges. As a result, we adopt the $\ell_1$-norm, its optimal convex approximation (Candès et al., 2006), to achieve similar sparsity with tractable computational demands. Specifically, we apply the $\ell_1$-norm to each row of $W^*$, thereby promoting sparsity throughout the matrix (see Appendix A for detailed explanations):

$$\left\| W_{i,:}^* \right\|_1, \ i = 1, 2, \ldots, m, \tag{2}$$

where $W_{i,:}^*$ represents the $i$-th row of $W^*$. Then, we construct our optimization model by integrating Equation 1 and Equation 2

$$\min_{W^* \in \mathbb{R}^{m \times n}} \frac{1}{2} \| W^* X^* - W X \|_F^2 + \lambda \sum_{i=1}^m \| W_{i,:}^* \|_1. \tag{3}$$

This model aims to simultaneously minimize both the output error and the sum of the $\ell_1$-norm values while the hyperparameter $\lambda > 0$ balances these two terms.

**Remark 1.** *The proposed optimization model in Equation 3 is convex. This is due to the fact that the square of the Frobenius norm is a convex function, as is the $\ell_1$-norm. Thus, the objective function, being a sum of these two convex functions, is also convex. Since the problem is an unconstrained optimization with a convex objective function, the overall optimization model is convex.*

### 3.2 OPTIMIZATION BASED ON FISTA

To deal with the non-smooth regularization term in Equation 3, a straightforward approach is using sub-gradient descent methods (Beck, 2017). However, its slow convergence rate of $\mathcal{O}(1/\sqrt{k})$ is not desirable. We thus turn to FISTA (Beck & Teboulle, 2009) with convergence rate $\mathcal{O}(1/k^2)$ to solve the proposed model Equation 3 efficiently. Specifically, starting with $t_0 = 1$ and an initial $W_0^*$, the $k$-th iteration of FISTA reads:

FISTA for Equation 3
$$\begin{cases} W_{k+\frac{1}{3}}^* = W_k^* - \frac{1}{L}\left( W_k^* X (X^*)^\top - W X (X^*)^\top \right), & \text{(4a)} \\[2mm] W_{k+\frac{2}{3}}^* = \text{SoftShrinkage}_{\frac{\lambda}{L}}\left( W_{k+\frac{1}{3}}^* \right), & \text{(4b)} \\[2mm] t_{k+1} = \frac{1}{2}\left( 1 + \sqrt{1 + 4t_k^2} \right), & \text{(4c)} \\[2mm] W_{k+1}^* = W_{k+\frac{2}{3}}^* + \frac{t_k - 1}{t_{k+1}}\left( W_{k+\frac{2}{3}}^* - W_k^* \right), & \text{(4d)} \end{cases}$$

where $L = \|\boldsymbol{X}^*(\boldsymbol{X}^*)^\top\|_2$ is the maximum eigenvalue of $\boldsymbol{X}^*(\boldsymbol{X}^*)^\top$ and the SoftShrinkage$_\rho(\cdot)$ operator with parameter $\rho \geq 0$ on a matrix $\boldsymbol{X} = (x_{ij}) \in \mathbb{R}^{m \times n}$ performs elementwise transformations defined by

$$\text{SoftShrinkage}_\rho(\boldsymbol{X}) = \boldsymbol{X}', \text{ where } x'_{ij} = \begin{cases} x_{ij} - \rho, & \text{if } x_{ij} > \rho, \\ x_{ij} + \rho, & \text{if } x_{ij} < -\rho, \\ x_{ij} = 0, & \text{otherwise.} \end{cases}$$

Step Equation 4a executes a gradient descent update on the parameter $\boldsymbol{W}_k^*$, aiming to minimize the function $1/2\|\boldsymbol{W}_k^*\boldsymbol{X}^* - \boldsymbol{W}\boldsymbol{X}\|_F^2$ with a step size of $1/L$. Step Equation 4b does a proximal update, defined as:

$$\boldsymbol{W}_{k+\frac{2}{3}}^* = \arg\min_{\boldsymbol{W}^*} \left\{ \frac{L}{2} \left\| \boldsymbol{W}^* - \boldsymbol{W}_{k+\frac{1}{3}}^* \right\|_F^2 + \lambda \sum_{i=1}^m \left\| \boldsymbol{W}_{i,:}^* \right\|_1 \right\}. \tag{5}$$

Steps Equation 4c and Equation 4d calculate a linear combination of the previous two points, $\left\{ \boldsymbol{W}_{k+\frac{2}{3}}^*, \boldsymbol{W}_k^* \right\}$, to facilitate accelerated convergence. Detailed derivations of these steps are provided in Appendix B. The FISTA iteration terminates either when the maximum number of iterations, $K$, is reached or when the following stopping criterion is satisfied:

$$\|\boldsymbol{W}_k^* - \boldsymbol{W}_{k-1}^*\|_F < 1 \times 10^{-6}. \tag{6}$$

### 3.3 EXTENSION TO 2:4 SEMI-STRUCTURED PRUNING

While our convex optimization framework effectively addresses unstructured pruning, practical deployment often necessitates structured or semi-structured sparsity patterns to fully leverage hardware acceleration capabilities. One notable pattern is the 2:4 semi-structured sparsity, which is supported by NVIDIA's Ampere architecture (Mishra et al., 2021), enabling significant speedups in inference.

The inclusion of the $n : m$ sparsity constraint render the optimization problem non-convex due to the combinatorial nature of selecting which elements to prune within each group. To tackle this challenge, we adopt FISTA updates, incorporating a hard thresholding step as follows:

$$\boldsymbol{W}_{K+1}^* = \mathcal{H}\left(\boldsymbol{W}_K^*, n : m\right), \tag{7}$$

where $W_K^*$ denote result from the $K$-th iteration of FISTA satisfying the stopping criterion, and $\mathcal{H}(\cdot)$ is the hard thresholding, which, for each group of four consecutive elements in every row, sets the two elements with the smallest absolute values to zero and retains the other two.

We acknowledge that the non-convex nature of this extension introduces complexities in theoretical analysis. However, the empirical success observed in our experiments provides confidence in the practical applicability of our approach.

### 3.4 ADAPTIVE HYPERPARAMETER TUNING

In Equation 3, the regularization parameter $\lambda$ plays a pivotal role in balancing the trade-off between the output error and the sparsity of the pruned weights $W^*$. A larger $\lambda$ emphasizes sparsity, potentially increasing the output error, while a smaller $\lambda$ focuses on minimizing the output error, resulting in less sparsity. To attain a specific desired sparsity level, it is essential to select an appropriate value of $\lambda$ that guides the optimization toward the target sparsity.

To automate the selection of $\lambda$, we propose employing an adaptive hyperparameter tuning mechanism based on the bisection method. This method iteratively adjusts $\lambda$ within a predefined interval $[0, M]$, where $M$ is a sufficiently large upper bound, to find the optimal value that yields the target sparsity upon solving the optimization problem using FISTA. We establish theoretical guarantees for the convergence of this method in the context of unstructured pruning, as stated in the following theorem:

**Theorem 1.** *Define $s(\lambda)$ as the function that maps the regularization parameter $\lambda$ to the sparsity level obtained after resolving the optimization problem. Let $s$ denote the desired sparsity level. The adaptive hyperparameter tuning mechanism leveraging the bisection method is guaranteed to converge to a $\lambda^*$ such that the resultant sparsity level $s(\lambda^*)$ satisfies the inequality $|s(\lambda^*) - s| \leq \epsilon$, where $\epsilon$ is a predefined tolerance.*

---

**Algorithm 1** FISTAPruner

**Inputs:** original output $\boldsymbol{WX}$, input activation $\boldsymbol{X}^*$, $\boldsymbol{W}_0^*$, $\lambda$, $K$, $T$, $\epsilon$, $s\%$ or $n:m$
$t \leftarrow 0;\ \ \boldsymbol{W}_{\text{best}}^* \leftarrow \boldsymbol{W}_0^*;\ \ \mathcal{E}_{\text{best}} \leftarrow \|\boldsymbol{W}_0^*\boldsymbol{X}^* - \boldsymbol{WX}\|_F$
**repeat**
    $\boldsymbol{W}_K^* \leftarrow \text{FISTA}\left(\boldsymbol{WX}, \boldsymbol{X}^*, \lambda, \boldsymbol{W}_{\text{best}}^*, K\right)$              # FISTA iterations as in Section 3.2
    $\boldsymbol{W}_{K+1}^* \leftarrow \mathcal{H}\left(\boldsymbol{W}_K^*, s\% \text{ or } n:m\right)$            # step for the specific sparsity pattern
    $\mathcal{E}_{\text{total}} \leftarrow \|\boldsymbol{W}_{K+1}^*\boldsymbol{X}^* - \boldsymbol{WX}\|_F$                # compute the total error
    $\mathcal{E}_{\text{round}} \leftarrow \mathcal{E}_{\text{total}} - \|\boldsymbol{W}_K^*\boldsymbol{X}^* - \boldsymbol{WX}\|_F$           # compute the rounding error
    **if** $\mathcal{E}_{\text{total}} < \mathcal{E}_{\text{best}}$ **then**
        $\boldsymbol{W}_{\text{best}}^* \leftarrow \boldsymbol{W}_{K+1}^*$                  # preserve the best solution
        $\mathcal{E}_{\text{stop}} = (\mathcal{E}_{\text{best}} - \mathcal{E}_{\text{total}})/\mathcal{E}_{\text{best}}$        # compute the stop condition
        $\mathcal{E}_{\text{best}} \leftarrow \mathcal{E}_{\text{total}}$                   # update the best total error
    **else**
        $t \leftarrow t + 1$           # update the number of steps without improvement
    **end if**
    update $\lambda$ by bisection based on $\mathcal{E}_{\text{round}}/\mathcal{E}_{\text{total}}$ as in Section 3.4
**until** $t \geq T$ **or** $\mathcal{E}_{\text{stop}} < \epsilon$
**return** $\boldsymbol{W}_{\text{best}}^*$

---

The proof is detailed in Appendix C. Although the adaptive hyperparameter tuning effectively identifies a regularization parameter $\lambda^*$ that yields a sparsity level close to the desired one, it may not always achieve the exact target due to the inherent continuous nature of the optimization process and limitations in numerical precision. To precisely attain the desired unstructured sparsity, we also implement a final hard thresholding step similar to Equation 7: after obtaining the optimized weights, the smallest-magnitude weights to zero until the exact sparsity level is achieved. To adjust $\lambda$ considering this hard thresholding step, we define the total error $\mathcal{E}_{\text{total}}$ and the rounding error $\mathcal{E}_{\text{round}}$ as

$$\mathcal{E}_{\text{total}} := \|\boldsymbol{W}_{K+1}^*\boldsymbol{X}^* - \boldsymbol{WX}\|_F, \ \mathcal{E}_{\text{round}} := \mathcal{E}_{\text{total}} - \|\boldsymbol{W}_K^*\boldsymbol{X}^* - \boldsymbol{WX}\|_F. \tag{8}$$

Building on the previous analysis, a high $\mathcal{E}_{\text{round}}/\mathcal{E}_{\text{total}}$ suggests that the majority of the error originates from the hard thresholding step. This suggests that the sparsity level of $W_K$ achieved via FISTA falls short of the desired sparsity, implying a need to increase the value of $\lambda$ to enhance the emphasis on the $\ell_1$-norm in Equation 3. Conversely, a low $\mathcal{E}_{\text{round}}/\mathcal{E}_{\text{total}}$ indicates that the sparsity in $W_K^*$ is adequate. This observation implies that a reduction in $\lambda$ might be beneficial. Such an adjustment would shift the model's emphasis towards minimizing output errors, thereby potentially decreasing the total error. Incorporating the above insights, we apply a threshold $\xi$ for $\mathcal{E}_{\text{round}}/\mathcal{E}_{\text{total}}$

### 3.5 FISTAPRUNER PSEUDOCODE

While the intra-layer error correction mechanism requires sequential pruning of the operators within a decoder layer, we could treat each decoder layer as an independent pruning unit, enabling parallel pruning across multiple decoder layers on different devices, which significantly enhances the efficiency. Within each decoder layer, the proposed FISTAPruner sequentially prune weights to eliminate error accumulations, as detailed in Section 3.1. Algorithm 1 presents FISTAPruner for the dense weight matrix $W$. It leverages FISTA to generate candidate sparse weights based on the model Equation 3, as detailed in Section 3.2. It then applies a hard thresholding step to meet specified sparsity constraints. Additionally, the parameter $\lambda$ is adaptively tuned, as detailed in Section 3.4, to optimize the trade-off between output error and sparsity. The algorithm iteratively updates the weights, preserving the best solution $W_{\text{best}}^*$, based on the lowest total error $\mathcal{E}_{\text{total}}$. It terminates when the number of consecutive iterations without an improvement in $W_{\text{best}}^*$ reaches $T$, or when the improvement ratio $(\mathcal{E}_{\text{best}} - \mathcal{E}_{\text{total}})/\mathcal{E}_{\text{best}}$ falls below the threshold $\epsilon$.

## 4 EXPERIMENTS

In this section, we detail a comprehensive set of experiments designed to validate the efficacy of FISTAPruner. We begin with an in-depth review of our experimental setup. Following this, we explore the perplexity and zero-shot capabilities of the pruned LLMs through rigorous testing and a series of ablation studies. Due to page length constraints, a portion of the results are presented in Appendix D.1, D.2 and D.3.

### 4.1 SETTINGS

**Models.** We utilize models from the OPT (Zhang et al., 2022), LLaMA (Touvron et al., 2023a), LLaMA-2 (Touvron et al., 2023b), and LLaMA-3 (Meta AI, 2023) families. Specifically, we assess our method across OPT-125M/350M/1.3B/2.7B/6.7B/13B/30B, LLaMA-7B/13B/30B/65B, LLaMA-2-7B/13B/70B, and LLaMA-3-8B/70B models.

**Benchmarks.** Our primary assessment focuses on evaluating the perplexity of pruned LLMs, a metric renowned for its reliability in assessing LLM performance. Following methodologies from previous studies (Frantar & Alistarh, 2023; Sun et al., 2023), we measure model perplexity using the WikiText-2-raw (Merity et al., 2016) (hereafter shortened to WikiText), PTB (Marcus et al., 1994), and C4 (Raffel et al., 2020) datasets. Additionally, we perform a comprehensive evaluation of the zero-shot capabilities of pruned LLaMA-3-70B models using several standard common-sense benchmark datasets. These include ARC Easy and ARC Challenge (Clark et al., 2018), WinoGrande (Sakaguchi et al., 2021), BoolQ (Clark et al., 2019), RTE (Wang et al., 2018), QNLI (Wang et al., 2018), and WNLI (Wang et al., 2018) tasks, facilitated by the LM Harness library (Gao et al., 2021).

**Baselines.** We compare FISTAPruner against two state-of-the-art pruning methods: SparseGPT (Frantar & Alistarh, 2023) and Wanda (Sun et al., 2023). Additionally, we evaluate against the latest training-free approach, DSnoT (Zhang et al., 2023d), which updates the results of other pruning methods, and the recent efficient prune-retrain approach, PERP (Zimmer et al., 2023). We evaluate two types of sparsity configurations: unstructured and 2:4 semi-structured sparsity.

**Setup.** We implement FISTAPruner using PyTorch (Paszke et al., 2019) and leverage the HuggingFace Transformers library (Wolf et al., 2019) for model and dataset management. All pruning experiments are conducted on NVIDIA A100 GPUs, each equipped with 80GB of memory. We observe that FISTAPruner efficiently prunes all LLMs using a single GPU and no more than 40GB of memory. For calibration data, we adhere to the approach outlined in previous works (Frantar & Alistarh, 2023; Sun et al., 2023), utilizing 128 sequences. Each sequence is composed of tokens sampled from the first shard of the C4 dataset, with the number of tokens equal to the maximum embedding length of the LLMs. For parameters of FISTAPruner, we set the initial value of $\lambda$ to $1 \times 10^{-5}$, $K$ to 20, $T$ to 3, $M$ to $10^6$, and $\xi$ to 0.3. For the OPT model family, we use the result of SparseGPT as a warm start for the FISTA iteration and set $\epsilon$ to $1 \times 10^{-6}$. For the LLaMA model family, we use the result of Wanda as a warm start and set $\epsilon$ to $1 \times 10^{-3}$.

### 4.2 PERPLEXITY EXPERIMENT RESULTS

In Tables 1 and 2, we present the perplexity results for the pruned OPT, LLaMA, LLaMA-2, and LLaMA-3 models of various sizes on WikiText. For results on PTB and C4, please refer to Appendix D.1 and D.2. We achieved a 50% unstructured or 2:4 semi-structured sparsity level by pruning all linear operators, excluding embeddings and the model head. The data in Tables 1 and 2 illustrate consistent improvements with FISTAPruner over SparseGPT and Wanda.

In Tables 3, we detail the comparison between FISTAPruner and DSnoT on LLaMA, LLaMA-2, and LLaMA-3 models of various sizes on WikiText. The data consistently indicate that FISTAPruner achieves lower perplexity scores, thereby surpassing DSnoT in performance.

We also compare FISTAPruner with the prune-retrain method PERP, with results presented in Table 4. These results demonstrate that FISTAPruner, without any retraining, outperforms the results of

Table 1: WikiText perplexity ($\downarrow$) of pruned OPT models under 50% unstructured and 2:4 semi-structured sparsity. FISTAPruner outperforms state-of-the-art methods.

| Method | Sparsity | OPT | | | | | | |
| | | 125M | 350M | 1.3B | 2.7B | 6.7B | 13B | 30B |
|---|---|---|---|---|---|---|---|---|
| Dense | 0% | 27.66 | 22.00 | 14.63 | 12.47 | 10.86 | 10.13 | 9.56 |
| SparseGPT | 50% | 37.01 | 31.53 | 17.55 | 13.46 | 11.60 | 11.15 | 9.77 |
| Wanda | 50% | 38.96 | 36.22 | 18.41 | 14.22 | 11.98 | 11.93 | 10.03 |
| FISTAPruner | 50% | **33.54** | **28.89** | **17.21** | **13.22** | **11.36** | **10.95** | **9.71** |
| SparseGPT | 2:4 | 60.02 | 50.15 | 23.83 | 17.20 | 14.13 | 12.94 | 10.92 |
| Wanda | 2:4 | 80.32 | 113.00 | 28.25 | 21.25 | 15.90 | 15.56 | 13.40 |
| FISTAPruner | 2:4 | **45.16** | **40.41** | **22.46** | **15.70** | **13.16** | **12.21** | **10.54** |

Table 2: WikiText perplexity (↓) of pruned LLaMA, LLaMA-2 and LLaMA-3 models under 50% unstructured and 2:4 semi-structured sparsity. FISTAPruner outperforms state-of-the-art methods.

| Method | Sparsity | LLaMA | | | | LLaMA-2 | | | LLaMA-3 | |
|---|---|---|---|---|---|---|---|---|---|---|
| | | 7B | 13B | 30B | 65B | 7B | 13B | 70B | 8B | 70B |
| Dense | 0% | 5.68 | 5.09 | 4.10 | 3.53 | 5.12 | 4.57 | 3.12 | 5.54 | 2.59 |
| SparseGPT | 50% | 7.24 | 6.22 | 5.33 | 4.60 | 6.54 | 5.63 | 3.99 | 8.64 | 5.30 |
| Wanda | 50% | 7.26 | 6.15 | 5.25 | 4.60 | 6.46 | 5.58 | 3.97 | 9.06 | 5.33 |
| FISTAPruner | 50% | **6.97** | **6.06** | **5.09** | **4.39** | **6.35** | **5.47** | **3.93** | **8.00** | **5.09** |
| SparseGPT | 2:4 | 11.32 | 9.11 | 7.21 | 6.24 | 10.37 | 8.29 | 5.38 | 14.65 | 8.63 |
| Wanda | 2:4 | 11.54 | 9.61 | 6.91 | 6.24 | 11.34 | 8.35 | 5.20 | 22.56 | 8.34 |
| FISTAPruner | 2:4 | **9.82** | **8.27** | **6.70** | **5.82** | **9.63** | **7.69** | **5.16** | **14.54** | **7.55** |

Table 3: WikiText perplexity (↓) of pruned LLaMA, LLaMA-2 and LLaMA-3 models under 50% unstructured and 2:4 semi-structured sparsity. FISTAPruner outperforms DSnoT.

| Method | Sparsity | LLaMA | | | LLaMA-2 | | LLaMA-3 |
|---|---|---|---|---|---|---|---|
| | | 7B | 13B | 30B | 7B | 13B | 8B |
| Wanda + DSnoT | 50% | 7.12 | 6.16 | 5.20 | 6.49 | 5.57 | 9.07 |
| FISTAPruner | 50% | **6.97** | **6.06** | **5.09** | **6.35** | **5.47** | **8.00** |
| Wanda + DSnoT | 2:4 | 11.54 | 9.49 | 7.09 | 11.53 | 8.52 | 20.56 |
| FISTAPruner | 2:4 | **9.82** | **8.27** | **6.70** | **9.63** | **7.69** | **14.54** |

Table 4: WikiText perplexity (↓) of pruned OPT models under 50% sparsity. FISTAPruner outperforms prune-retrain approach PERP.

| Method | Sparsity | OPT-2.7B | OPT-6.7B | OPT-13B | OPT-30B |
|---|---|---|---|---|---|
| SparseGPT + PERP | 50% | 13.40 | 11.47 | 10.85 | 9.76 |
| Wanda + PERP | 50% | 13.88 | 11.83 | 11.06 | 10.04 |
| FISTAPruner | 50% | **13.22** | **11.36** | **10.95** | **9.71** |

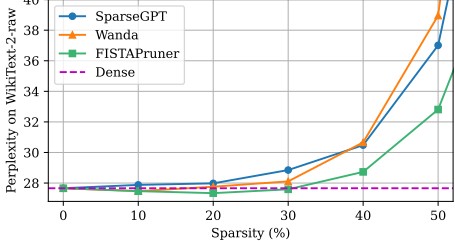

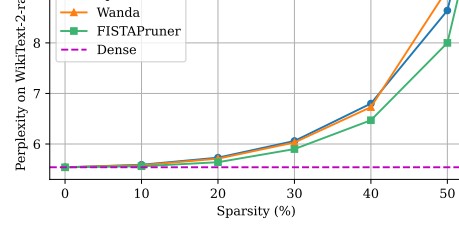

(a) Perplexity-vs-Sparsity on OPT-125M.  (b) Perplexity-vs-Sparsity on LLaMA-3-8B.

Figure 3: Comparative analysis of sparsity versus perplexity across different methods for OPT-125M and LLaMA-3-8B models on WikiText dataset.

SparGPT/Wanda retrained using PERP. Moreover, our method is also compatible with retraining methods and could serve as a superior initialization point in the retraining process.

To further investigate FISTAPruner's performance under different unstructured sparsity levels, we conducted experiments on the OPT-125M and LLaMA-3-8B models, with perplexity results visualized in Figure 3 and measured using WikiText. The results indicate that FISTAPruner consistently outperforms existing methods across different levels of unstructured sparsity. Notably, at 20% unstructured sparsity on the OPT-125M model, FISTAPruner's performance even surpasses that of the dense network.

### 4.3 ZERO-SHOT TASK RESULTS

The results of zero-shot tasks on pruned LLaMA-3-70B models, with 50% unstructured and 2:4 semi-structured sparsity, are detailed in Table 5. These results indicate that FISTAPruner surpasses existing methods on most tasks. Furthermore, when evaluating the average accuracy across the seven

tasks we examined, FISTAPruner consistently shows superior performance compared to existing methods, particularly with 2:4 semi-structured sparsity.

Table 5: Zero-shot results (accuracy, ↑) of the pruned LLaMA-3-70B model under 50% unstructured and 2:4 semi-structured sparsity. FISTAPruner outperforms state-of-the-art methods on most of the tasks and yields much higher average accuracies especially under 2:4 semi-structured sparsity.

| Method | Sparsity | ARC-c | ARC-e | WinoGrande | RTE | BoolQ | QNLI | WNLI | Mean |
|---|---|---|---|---|---|---|---|---|---|
| Dense | 0% | 0.6024 | 0.8685 | 0.8035 | 0.6859 | 0.8560 | 0.5190 | 0.7183 | 0.7219 |
| SparseGPT | 50% | 0.5401 | 0.8340 | 0.7979 | 0.7040 | 0.8480 | 0.5035 | 0.7042 | 0.7045 |
| Wanda | 50% | 0.5427 | 0.8320 | 0.7814 | **0.7076** | 0.8480 | 0.5045 | 0.6338 | 0.6928 |
| FISTAPruner | 50% | **0.5614** | **0.8410** | **0.8035** | 0.6895 | **0.8645** | **0.5055** | **0.7183** | **0.7120** |
| SparseGPT | 2:4 | 0.4590 | 0.7830 | 0.7609 | 0.6426 | 0.8165 | 0.4985 | 0.5493 | 0.6443 |
| Wanda | 2:4 | **0.4829** | 0.7860 | 0.7174 | 0.6354 | 0.7615 | 0.5390 | 0.6056 | 0.6468 |
| FISTAPruner | 2:4 | 0.4735 | **0.7985** | **0.7751** | **0.7004** | **0.8540** | **0.5675** | **0.6620** | **0.6901** |

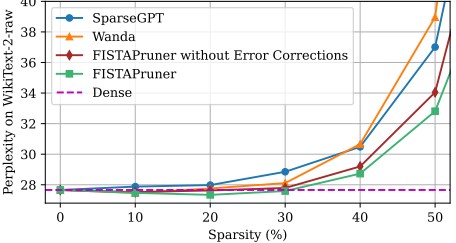

(a) Intra-layer error corrections ablation.

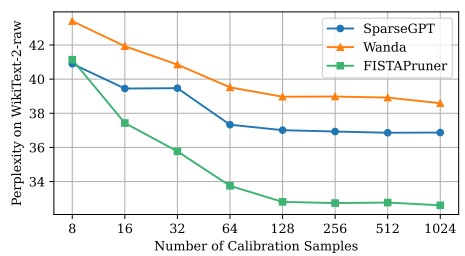

(b) Calibration samples ablation.

Figure 4: Studies of FISTAPruner on the WikiText dataset on OPT-2 125M, showcasing the effects of intra-layer error correction and varying calibration sample sizes.

### 4.4 ABLATION STUDY ON INTRA-LAYER ERROR CORRECTIONS

We perform ablation studies on the OPT-125M model with 50% unstructured sparsity to evaluate the intra-layer error correction mechanism. We compare the performance of FISTAPruner with and without the intra-layer error correction mechanism, with results on the WikiText dataset displayed in Figure 4(a) (see results on PTB and C4 datasets in Appendix D.3). We observe that the perplexity of the pruned model incorporating this mechanism consistently outperforms the version without it, thereby confirming its effectiveness. Moreover, FISTAPruner, even without the intra-layer error correction mechanism, outperforms existing methods such as SparseGPT and Wanda. This underscores the effectiveness of applying convex optimization theory and algorithms to pruning problems. Additionally, we treat each decoder layer as an independent pruning unit with intra-layer error correction, rather than using both intra- and inter-layer error correction for a global mechanism, for the following reasons: (1) Intra-layer error correction allows independent pruning of each decoder layer, enabling distribution of the task across multiple devices and improving overall efficiency. (2) While combining intra- and inter-layer error correction can reduce error accumulation, it is effective only at low sparsity levels. At higher sparsity, global error correction dominates layer-specific pruning, leading to worse performance. A detailed analysis of this is provided in Appendix E.

### 4.5 IMPACT OF CALIBRATION DATA AND WARM START

We conduct studies to evaluate the the impact of the number of calibration samples and warm start.

**Amount of Calibration Data.** We investigate the performance of FISTAPruner and existing methods, SparseGPT and Wanda, in relation to the number of calibration data samples, which we vary in powers of two. The results for the WikiText dataset with the OPT-125M model at 50% sparsity are shown in Figure 4(b) (additional results for the PTB and C4 datasets are provided in Appendix D.3). We observe that using more calibration samples significantly enhances performance, but only up to

a certain point as the improvement curve quickly flattens. This finding aligns with observations in (Frantar & Alistarh, 2023; Sun et al., 2023). Given that using more samples increases computational and memory costs, we consistently use 128 calibration samples in all our experiments.

**Warm Start.** Warm start is a widely recognized technique in optimization that leverages starting at a point near the optimal solution to significantly reduce the total convergence time. In our framework, we evaluate the efficiency of warm start mechanism as follows: Dense Weights $<$ Magnitude Pruning $\approx$ Wanda $<$ SparseGPT. Dense weights, though readily obtainable, slow down the convergence due to their significant deviation from the target sparsity level. Magnitude pruning, involving absolute value computations and comparisons, meets sparsity requirements but generally yields lower-quality solutions. Wanda, requiring absolute value computations, $\ell_2$-norm calculations of activation columns, and element-wise multiplication, is nearly as efficient as magnitude pruning. This near parity in efficiency is due to our model's reliance on activation data from calibration, allowing $\ell_2$-norm computations to occur incidentally during the process. Despite their similar efficiencies, Wanda's solutions markedly outperform those from magnitude pruning. SparseGPT is less efficient compared with magnitude pruning and Wanda but may provide a stronger initial point.

To further illustrate the impact of warm start on FISTAPruner, we conduct additional tests using both dense weights and magnitude pruning results as starting points. The results are presented in Table 6, which indicates that FISTAPruner still can achieve comparable results.

Table 6: Perplexity ($\downarrow$) results for WikiText, PTB and C4 under 50% unstructured and 2:4 semi-structured sparsity. FISTAPruner is initialized with magnitude pruning and dense weights.

| Method | Sparsity | WikiText | PTB | C4 |
|---|---|---|---|---|
| Magnitude | 25% | 31.38 | 38.99 | 26.56 |
| FISTAPruner (initialized with magnitude pruning) | 25% | 28.67 | 40.29 | 27.07 |
| FISTAPruner (initialized with dense weights) | 25% | **28.66** | **40.27** | 27.07 |
| Magnitude | 50% | 193.35 | 276.17 | 141.00 |
| FISTAPruner (initialized with magnitude pruning) | 50% | 38.62 | **52.26** | **32.87** |
| FISTAPruner (initialized with dense weights) | 50% | 38.62 | 52.43 | 32.89 |
| Magnitude | 2:4 | 343.91 | 810.42 | 223.98 |
| FISTAPruner (initialized with magnitude pruning) | 2:4 | **57.43** | **78.37** | **45.20** |
| FISTAPruner (initialized with dense weights) | 2:4 | 58.55 | 80.72 | 45.51 |

## 5 DISCUSSION

Despite the rigorous theoretical foundation and impressive pruning performance of FISTAPruner, the time required for pruning remains a limitation of our method compared to SparseGPT and Wanda. This is primarily due to the iterative nature of FISTA and the process of tuning $\lambda$. Pruning time varies with model size; for instance, it takes about 10 minutes for OPT-125M, while LLaMA-3-70B requires approximately 12 hours on a single Nvidia A100 GPU with 40GB of memory. However, the parallel-pruning capability of FISTAPruner, which allows for simultaneous pruning of multiple decoder layers across various devices, can mitigate this issue to some extent. Furthermore, as post-training pruning is typically an offline process, time sensitivity may not be a critical factor in real-world applications. In addition, FISTAPruner represents an attempt to integrate convex optimization theory and algorithms into LLM applications, potentially inspiring further advancements in this area.

## 6 CONCLUSION

In this paper, we introduce FISTAPruner, a layer-wise post-training pruning method for LLMs. Initially, we develop a convex optimization model that employs the $\ell_1$-norm to induce unstructured sparsity in the weights, complemented by an intra-layer error correction mechanism to eliminate cumulative errors across operators in the traditional pruning process. Subsequently, we utilize FISTA to efficiently solve the proposed model. Additionally, we extend FISTAPruner to accommodate $n : m$ semi-structured pruning. FISTAPruner supports parallel pruning, which can reduce the total pruning time by utilizing various devices simultaneously. Extensive experiments on the OPT, LLaMA, LLaMA-2, and LLaMA-3 model families demonstrate FISTAPruner's superior performance compared to existing methods. We hope this work enhances understanding of sparsity in LLMs and inspires further integration of convex optimization within LLM applications.

## REPRODUCIBILITY STATEMENT

We have taken several measures to ensure the reproducibility of our work. Detailed descriptions of our methods, including all hyperparameter settings and experimental configurations, are provided in Section 4.1. For theoretical results, all assumptions and proofs are thoroughly explained and included in the main text and appendix for clarity and verification. We will release our code upon acceptance.

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

## A   DERIVATIONS OF THE PROPOSED OPTIMIZATION MODEL

We present detailed derivations of Model Equation 3 in the following. Given $\boldsymbol{X}^* \in \mathbb{R}^{n \times p}$ and $\boldsymbol{W}\boldsymbol{X} \in \mathbb{R}^{m \times p}$, we want to find a sparse solution $W^* \in \mathbb{R}^{m \times n}$ that minimizes the pruning metric

$$\|\boldsymbol{W}^*\boldsymbol{X}^* - \boldsymbol{W}\boldsymbol{X}\|_F. \tag{9}$$

We observe its similarities to the well-known least absolute shrinkage and selection operator (LASSO) (Tibshirani, 1996) problem and thus transform it into a standard LASSO model, which could be efficiently solved by operator-splitting algorithms such as FISTA. To achieve such a transformation, first, we leverage the following equality to write the decision variable $W^*$ in its vector form:

$$
\begin{aligned}
\|\boldsymbol{W}^*\boldsymbol{X}^* - \boldsymbol{W}\boldsymbol{X}\|_F^2 &= \left\|(\boldsymbol{X}^*)^\top(\boldsymbol{W}^*)^\top - (\boldsymbol{W}\boldsymbol{X})^\top\right\|_F^2 \\
&= \sum_{i=1}^m \left\|(\boldsymbol{X}^*)^\top(\boldsymbol{W}_{i,:}^*)^\top - (\boldsymbol{W}\boldsymbol{X})_{i,:}^\top\right\|_2^2 \\
&= \left\| \begin{pmatrix} (\boldsymbol{X}^*)^\top & & \\ & \ddots & \\ & & (\boldsymbol{X}^*)^\top \end{pmatrix} \begin{pmatrix} (\boldsymbol{W}_{1,:}^*)^\top \\ (\boldsymbol{W}_{2,:}^*)^\top \\ \vdots \\ (\boldsymbol{W}_{m,:}^*)^\top \end{pmatrix} - \begin{pmatrix} (\boldsymbol{W}\boldsymbol{X})_{1,:}^\top \\ (\boldsymbol{W}\boldsymbol{X})_{2,:}^\top \\ \vdots \\ (\boldsymbol{W}\boldsymbol{X})_{m,:}^\top \end{pmatrix} \right\|_2^2
\end{aligned}
$$

Then we can rewrite the square of the pruning metric in its vector form,

$$\|\boldsymbol{A}\boldsymbol{x} - \boldsymbol{b}\|_2^2, \tag{10}$$

where

$$
\boldsymbol{A} = \begin{pmatrix} (\boldsymbol{X}^*)^\top & & \\ & \ddots & \\ & & (\boldsymbol{X}^*)^\top \end{pmatrix} \in \mathbb{R}^{pm \times nm}, \ \boldsymbol{x} = \begin{pmatrix} (\boldsymbol{W}_{1,:}^*)^\top \\ (\boldsymbol{W}_{2,:}^*)^\top \\ \vdots \\ (\boldsymbol{W}_{m,:}^*)^\top \end{pmatrix} \in \mathbb{R}^{nm}, \ \boldsymbol{b} = \begin{pmatrix} (\boldsymbol{W}\boldsymbol{X})_{1,:}^\top \\ (\boldsymbol{W}\boldsymbol{X})_{2,:}^\top \\ \vdots \\ (\boldsymbol{W}\boldsymbol{X})_{m,:}^\top \end{pmatrix} \in \mathbb{R}^{pm}.
$$

Note that finding a sparse $W^*$ to minimize Equation 9 is equivalent to finding a sparse $\mathbf{x}$ to minimize Equation 10, which could be modeled by the LASSO formulation

$$\min_{\mathbf{x}} \frac{1}{2}\|\boldsymbol{A}\boldsymbol{x} - \boldsymbol{b}\|_2^2 + \lambda\|\boldsymbol{x}\|_1.$$

Now, we have

$$
\begin{aligned}
\frac{1}{2}\|\boldsymbol{A}\boldsymbol{x} - \boldsymbol{b}\|_2^2 + \|\boldsymbol{x}\|_1 &= \frac{1}{2}\|\boldsymbol{W}^*\boldsymbol{X}^* - \boldsymbol{W}\boldsymbol{X}\|_F^2 + \lambda \left\| \begin{pmatrix} (\boldsymbol{W}_{1,:}^*)^\top \\ (\boldsymbol{W}_{2,:}^*)^\top \\ \vdots \\ (\boldsymbol{W}_{m,:}^*)^\top \end{pmatrix} \right\|_1 \\
&= \frac{1}{2}\|\boldsymbol{W}^*\boldsymbol{X}^* - \boldsymbol{W}\boldsymbol{X}\|_F^2 + \lambda \sum_{i=1}^m \left\|(\boldsymbol{W}_{i,:}^*)^\top\right\|_1,
\end{aligned}
$$

and hence, we obtain the proposed optimization model Equation 3.

## B   DERIVATIONS OF THE FISTA ITERATIONS

We derive here the FISTA Iterations for the optimization problem Equation 3 in which one full iteration includes a gradient descent step of the quadratic term $\frac{1}{2}\|\boldsymbol{W}^*\boldsymbol{X}^* - \boldsymbol{W}\boldsymbol{X}\|_F^2$, a proximal step of the regularization term $\lambda \sum_{i=1}^m \left\|(\boldsymbol{W}_{i,:}^*)^\top\right\|_1$ and a Nestrov acceleration term that yields a improved convergence rate of $O(1/k^2)$ (Beck & Teboulle, 2009).

Let $f : \mathbb{R}^{m \times n} \to \mathbb{R}_+$ be a function defined by

$$f(\boldsymbol{Y}) := \frac{1}{2}\|\boldsymbol{Y}\boldsymbol{X}^* - \boldsymbol{W}\boldsymbol{X}\|_F^2.$$

The gradient of $f$ at $Y = W_k^*$ is computed as

$$\nabla f(W_k^*) = (W_k^* X^* - WX)(X^*)^\top$$
$$= W_k^* X^* (X^*)^\top - WX(X^*)^\top.$$

Thus, given optimal step size $1/L$ where $L$ is the maximum eigenvalue of $X^*(X^*)^\top$ (Beck & Teboulle, 2009), the gradient descent step Equation 4a of FISTA reads as

$$W_{k+\frac{1}{3}}^* = W_k^* - \frac{1}{L}\left(W_k^* X(X^*)^\top - WX(X^*)^\top\right).$$

In the second step Equation 4b, we do a proximal update with respect to the regularization term by solving

$$\min_{W^* \in \mathbb{R}^{m \times n}} \frac{L}{2}\left\|W^* - W_{k+\frac{1}{3}}^*\right\|_F^2 + \lambda \sum_{i=1}^m \|W_{i,:}^*\|_1. \tag{11}$$

Let $h : \mathbb{R} \to \mathbb{R}_+$ be a function defined by

$$h(y|z) := \frac{1}{2}(y - z)^2 + \frac{\lambda}{L}|y|.$$

Observe that

$$\frac{L}{2}\left\|W^* - W_{k+\frac{1}{3}}^*\right\|_F^2 + \lambda \sum_{i=1}^m \|W_{i,:}^*\|_1 = L \sum_{i,j} h\left(W_{ij}^* \middle| W_{k+\frac{1}{3},ij}^*\right).$$

Hence problem Equation 11 can be split into $m \times n$ independent subproblems of dimension 1 and we only need to focus on solving each one of them. Note that $h$ is convex but not smooth. It suffices to find a point $W_{k+\frac{2}{3},ij}^*$ such that

$$0 \in \partial h\left(W_{k+\frac{2}{3},ij}^* \middle| W_{k+\frac{1}{3},ij}^*\right),$$

where $\partial$ denotes the sub-differential operator. Observe that

$$\partial h(y|z) = \begin{cases} y - z + \frac{\lambda}{L}, & \text{if } y > 0, \\ y - z - \frac{\lambda}{L}, & \text{if } y < 0, \\ \{y - z + u\frac{\lambda}{L} \mid u \in [-1, 1]\}, & \text{if } y = 0. \end{cases}$$

We now solve for $0 \in \partial h(y|z)$ by considering the following cases:

- If $y > 0$, then we set $y - z + \frac{\lambda}{L} = 0$. This gives $y = z - \frac{\lambda}{L}$ and requires $z > \frac{\lambda}{L}$.
- If $y < 0$, then we set $y - z - \frac{\lambda}{L} = 0$. This gives $y = z + \frac{\lambda}{L}$ and requires $z < -\frac{\lambda}{L}$.
- If $y = 0$, then we want $0 \in \{y - z + u\frac{\lambda}{L} \mid u \in [-1, 1]\}$. This requires $-\frac{\lambda}{L} < z < \frac{\lambda}{L}$.

Hence, $0 \in \partial h\left(W_{k+\frac{2}{3},ij}^* \middle| W_{k+\frac{1}{3},ij}^*\right)$ yields

$$W_{k+\frac{2}{3},ij}^* = \begin{cases} W_{k+\frac{1}{3},ij}^* - \frac{\lambda}{L}, & \text{if } W_{k+\frac{1}{3},ij}^* > \frac{\lambda}{L}, \\ W_{k+\frac{1}{3},ij}^* + \frac{\lambda}{L}, & \text{if } W_{k+\frac{1}{3},ij}^* < -\frac{\lambda}{L}, \\ 0, & \text{otherwise}, \end{cases}$$

which is exactly the value given by $\text{SoftShrinkage}_{\lambda/L}\left(W_{k+\frac{1}{3},ij}^*\right)$.

Finally, according to (Beck & Teboulle, 2009), we add a Nestrov acceleration step by setting $t_0 = 1$ and computing

$$t_{k+1} = \frac{1}{2}\left(1 + \sqrt{1 + 4t_k^2}\right), \tag{12}$$

$$W_{k+1}^* = W_{k+\frac{2}{3}}^* + \frac{t_k - 1}{t_{k+1}}\left(W_{k+\frac{2}{3}}^* - W_k^*\right), \tag{13}$$

which gives steps Equation 4c and Equation 4d.

The above illustrates the details of the FISTA iterations.

## C  PROOF OF THEOREM 1

The function $s(\lambda)$ is continuous because small changes in $\lambda$ lead to small changes in the sparsity level due to the continuity of the optimization problem's solution with respect to $\lambda$. Besides, $s(\lambda)$ is non-decreasing because increasing $\lambda$ enhances the sparsity-promoting effect of the $\ell_1$-norm, potentially resulting in more zero entries in $\boldsymbol{W}^*$. Therefore, $s(\lambda)$ is a continuous and non-decreasing function of $\lambda$ over the interval $[0, M]$ for the bisection method. At $\lambda = 0$, the problem reduces to minimizing the reconstruction error without regularization, yielding the least sparse solution ($s(0) = 0\%$ sparsity). At a sufficiently large $\lambda = M$, the regularization term dominates, driving most weights to zero ($s(M) = 100\%$ sparsity). Since $s(\lambda)$ is continuous and $s(0) \leq s \leq s(M)$, the Intermediate Value Theorem guarantees the existence of $\lambda^* \in [0, M]$ such that $s(\lambda^*) = s$. The bisection method halves the interval $[\lambda_{\min}, \lambda_{\max}]$ in each iteration, ensuring convergence to $\lambda^*$ within a tolerance $\delta$ after $k \geq \log_2\left(\frac{\lambda_{\max} - \lambda_{\min}}{\delta}\right)$ iterations. Since $s(\lambda)$ is continuous and non-decreasing, the sparsity level $s_k$ achieved at each iteration converges to $s$. The bisection methods terminates when $|s_k - s| \leq \epsilon$, ensuring that the final $\lambda^* = \lambda_k$ yields a sparsity level within the desired tolerance.

## D  ADDITIONAL RESULTS

### D.1  PERPLEXITY RESULTS ON PTB

We present the PTB perplexity results of pruned OPT, LLaMA, LLaMA-2, and LLaMA-3 models under 50% unstructured and 2:4 semi-structured sparsity in Tables 7 and 8. FISTAPruner outperforms state-of-the-art methods on all OPT, LLaMA and LLaMA-3 models, as well as on most LLaMA-2 models on the PTB dataset. The sole exception is the pruning of the LLaMA-2-70B model under 50% unstructured sparsity, where FISTAPruner surpasses Wanda but falls short of SparseGPT. This underperformance may be due to the generally poorer performance of LLaMA-2 models compared to similarly sized models from other families. For instance, the dense LLaMA-2-13B model exhibits a PTB perplexity of 56.52, even higher than the smaller LLaMA-2-7B model, which has a perplexity of 50.19. Moreover, we observe that the PTB perplexity results for all dense LLaMA and LLaMA-2 models are consistently higher than those for similarly sized OPT models; for example, the LLaMA-2-13B's perplexity of 56.52 far exceeds the smallest OPT-125M model's 38.99. In contrast, LLaMA-3 models show significantly better performance on the PTB dataset.

Table 7: PTB perplexity of pruned OPT models under 50% unstructured and 2:4 semi-structured sparsity. FISTAPruner outperforms state-of-the-art methods.

| | | OPT | | | | | | |
|---|---|---|---|---|---|---|---|---|
| Method | Sparsity | 125M | 350M | 1.3B | 2.7B | 6.7B | 13B | 30B |
| Dense | 0% | 38.99 | 31.07 | 20.29 | 17.97 | 15.77 | 14.52 | 14.04 |
| SparseGPT | 50% | 55.38 | 43.58 | 25.64 | 20.52 | 17.38 | 15.98 | 14.97 |
| Wanda | 50% | 57.60 | 55.47 | 27.98 | 21.85 | 17.92 | 17.45 | 15.47 |
| FISTAPruner | 50% | **49.79** | **41.26** | **25.08** | **20.15** | **17.08** | **15.87** | **14.92** |
| SparseGPT | 2:4 | 94.21 | 72.82 | 37.30 | 26.87 | 21.65 | 18.69 | 16.56 |
| Wanda | 2:4 | 111.55 | 135.98 | 43.85 | 34.64 | 25.07 | 22.16 | 21.65 |
| FISTAPruner | 2:4 | **67.80** | **59.51** | **36.26** | **24.43** | **20.04** | **18.08** | **16.18** |

Table 8: PTB perplexity ($\downarrow$) of pruned LLaMA, LLaMA-2 and LLaMA-3 models under 50% unstructured and 2:4 semi-structured sparsity.

| | | LLaMA | | | | LLaMA-2 | | | LLaMA-3 | |
|---|---|---|---|---|---|---|---|---|---|---|
| Method | Sparsity | 7B | 13B | 30B | 65B | 7B | 13B | 70B | 8B | 70B |
| Dense | 0% | 41.15 | 28.10 | 23.51 | 25.07 | 50.19 | 56.52 | 22.68 | 10.17 | 7.87 |
| SparseGPT | 50% | 79.67 | 37.49 | 26.14 | 27.64 | 1020.01 | 95.41 | **24.87** | 14.00 | 9.24 |
| Wanda | 50% | 80.48 | 36.43 | 26.64 | 25.77 | 97.58 | 86.79 | 26.07 | 15.54 | 9.44 |
| FISTAPruner | 50% | **58.67** | **35.30** | **25.63** | **25.15** | **96.72** | **78.23** | 25.36 | **12.93** | **8.88** |
| SparseGPT | 2:4 | 154.62 | 71.68 | 32.44 | 32.91 | 1163.57 | 154.15 | 31.51 | 23.42 | 13.01 |
| Wanda | 2:4 | 211.40 | 74.29 | 35.56 | 33.39 | 587.54 | 224.55 | 33.97 | 48.96 | 14.17 |
| FISTAPruner | 2:4 | **91.84** | **64.04** | **30.86** | **30.78** | **361.16** | **136.84** | **31.49** | **22.60** | **11.11** |

## D.2 PERPLEXITY RESULTS ON C4

The C4 perplexity results of pruned OPT, LLaMA, LLaMA-2, and LLaMA-3 models under 50% unstructured and 2:4 semi-structured sparsity are shown in Tables 7 and 8. FISTAPruner performs consistently better than the state-of-the-art methods.

Table 9: C4 perplexity (↓) of pruned OPT models under 50% unstructured and 2:4 semi-structured sparsity. FISTAPruner outperforms state-of-the-art methods.

| | | OPT | | | | | | |
|---|---|---|---|---|---|---|---|---|
| Method | Sparsity | 125M | 350M | 1.3B | 2.7B | 6.7B | 13B | 30B |
| Dense | 0% | 26.56 | 22.59 | 16.07 | 14.34 | 12.71 | 12.06 | 11.45 |
| SparseGPT | 50% | 33.52 | 29.14 | 19.23 | 15.77 | 13.73 | 12.98 | 11.96 |
| Wanda | 50% | 34.89 | 34.46 | 20.63 | 16.44 | 14.25 | 13.57 | 12.32 |
| FISTAPruner | 50% | **30.93** | **27.36** | **18.56** | **15.58** | **13.61** | **12.94** | **11.92** |
| SparseGPT | 2:4 | 52.11 | 46.36 | 25.77 | 19.35 | 16.44 | 14.85 | 13.18 |
| Wanda | 2:4 | 64.73 | 88.62 | 28.59 | 22.88 | 19.00 | 16.19 | 16.18 |
| FISTAPruner | 2:4 | **38.08** | **36.45** | **24.29** | **17.82** | **15.35** | **14.19** | **12.78** |

Table 10: C4 perplexity (↓) of pruned LLaMA, LLaMA-2 and LLaMA-3 models under 50% unstructured and 2:4 semi-structured sparsity. FISTAPruner outperforms state-of-the-art methods.

| | | LLaMA | | | | LLaMA-2 | | | LLaMA-3 | |
|---|---|---|---|---|---|---|---|---|---|---|
| Method | Sparsity | 7B | 13B | 30B | 65B | 7B | 13B | 70B | 8B | 70B |
| Dense | 0% | 7.34 | 6.80 | 6.13 | 5.81 | 7.04 | 6.52 | 5.53 | 9.01 | 6.82 |
| SparseGPT | 50% | 9.33 | 8.14 | 7.34 | 6.66 | 9.00 | 7.96 | 6.25 | 13.93 | 9.34 |
| Wanda | 50% | 9.34 | 8.15 | 7.29 | 6.71 | 8.94 | 8.04 | 6.30 | 14.97 | 9.80 |
| FISTAPruner | 50% | **8.90** | **7.96** | **7.05** | **6.49** | **8.62** | **7.73** | **6.22** | **13.12** | **8.94** |
| SparseGPT | 2:4 | 13.65 | 11.38 | 9.50 | 8.41 | 13.58 | 11.39 | 7.99 | 24.16 | 14.81 |
| Wanda | 2:4 | 14.47 | 12.11 | 9.46 | 8.78 | 15.07 | 12.13 | 7.89 | 36.70 | 14.47 |
| FISTAPruner | 2:4 | **11.95** | **10.27** | **8.81** | **7.82** | **12.41** | **10.34** | **7.59** | **23.15** | **12.18** |

## D.3 ADDITIONAL STUDY RESULTS

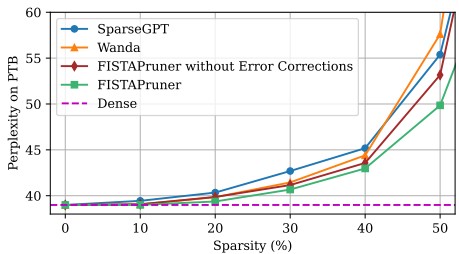 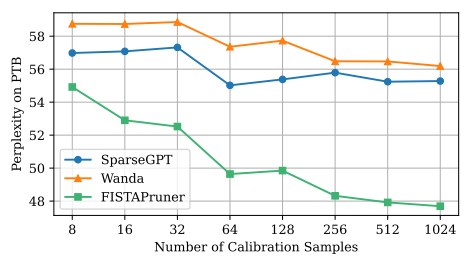

(a) Intra-layer error corrections ablation.     (b) Calibration samples ablation.

Figure 5: Studies of FISTAPruner on the PTB dataset on OPT-125M, showcasing the effects of intra-layer error correction and varying calibration sample sizes.

**Intra-layer Error Corrections.** We compare the performance of FISTAPruner with and without the intra-layer error correction mechanism, with results on PTB and C4 datasets displayed in Figures 5(a) and 6(a). The results indicate that the perplexity of the pruned model incorporating this mechanism consistently outperforms the version without it, thereby confirming its effectiveness.

**Amount of Calibration Data.** The results of pruning performance in relation to the number of calibration data samples on PTB and C4 datasets are displayed in Figures 5(b) and 6(b). The same curve pattern as shown in Figure 4(b) is observed.

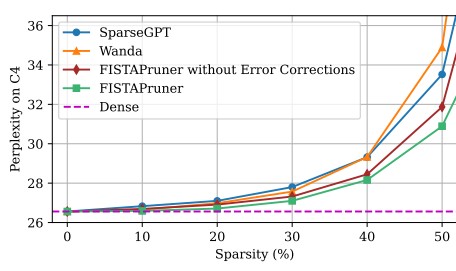 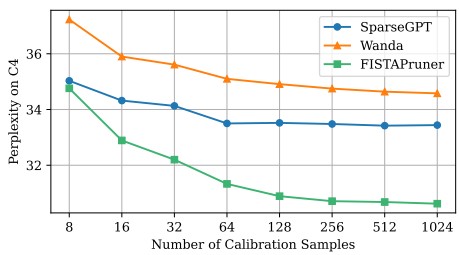

(a) Intra-layer error corrections ablation.

(b) Calibration samples ablation.

Figure 6: Studies of FISTAPruner on the C4 dataset on OPT-125M, showcasing the effects of intra-layer error correction and varying calibration sample sizes.

## E  WHY INTRA-LAYER ERROR CORRECTION IS PREFERRED OVER INTRA- AND INTER-LAYER ERROR CORRECTION

We apply only the intra-layer error correction mechanism for two reasons:

1. **Parallelization:** Intra-layer error correction enables independent pruning of each decoder layer, allowing us to distribute the pruning task across multiple devices by assigning different decoder layers to different devices. This increases the overall pruning efficiency.

2. **Sparsity Sensitivity:** While combining intra- and inter-layer error correction could intuitively reduce error accumulation across the network, we found that this approach is effective only at low sparsity levels. When the pruning task becomes harder (i.e., higher sparsity), global error correction tends to overshadow the pruning process of individual layers, ultimately leading to worse performance.

The first reason is straightforward; we will explain the second reason in more detail below.

We conducted a series of comparison experiments on OPT-125M at sparsity levels of 5%, 10%, 20%, and 50%. The experiments included three conditions: intra-layer error correction only, both intra- and inter-layer error correction, and no error correction. The results are presented in the following tables.

Table 11: OPT-125M under 5% Sparsity

|  | WikiText | C4 | PTB |
|---|---|---|---|
| Intra-layer Error Correction Only | 27.64 | 26.57 | 38.99 |
| Intra-layer and Inter-layer Error Correction | 27.63 | 26.56 | 38.98 |
| No Error Correction | 27.69 | 26.60 | 38.98 |

Table 12: OPT-125M under 10% Sparsity

|  | WikiText | C4 | PTB |
|---|---|---|---|
| Intra-layer Error Correction Only | 27.47 | 26.59 | 39.00 |
| Intra-layer and Inter-layer Error Correction | 27.43 | 26.58 | 39.04 |
| No Error Correction | 27.52 | 26.69 | 39.07 |

As shown in the results above, we summarize the perplexity comparison across different sparsity levels as follows:

- **5% and 10%:** intra- and inter-layer error correction < intra-layer error correction only < no error correction.

Table 13: OPT-125M under 20% Sparsity

|  | WikiText | C4 | PTB |
|---|---|---|---|
| Intra-layer Error Correction Only | 27.36 | 26.71 | 39.39 |
| Intra-layer and Inter-layer Error Correction | 27.37 | 26.72 | 39.53 |
| No Error Correction | 27.61 | 26.91 | 39.85 |

Table 14: OPT-125M under 50% Sparsity

|  | WikiText | C4 | PTB |
|---|---|---|---|
| Intra-layer Error Correction Only | 33.54 | 30.93 | 49.79 |
| Intra-layer and Inter-layer Error Correction | 35.90 | 32.93 | 55.24 |
| No Error Correction | 34.48 | 32.24 | 54.11 |

- **20%:** intra-layer error correction only $<$ intra- and inter-layer error correction $<$ no error correction.

- **50%:** intra-layer error correction only $<$ no error correction $<$ intra- and inter-layer error correction.

First, the results confirm the effectiveness of our intra-layer error correction mechanism, as it consistently outperforms the no-error-correction approach.

Second, the results confirm the effectiveness of using both intra- and inter-layer error correction at low sparsity levels, as it consistently outperforms the intra-layer error correction alone at 5% and 10% sparsity.

Third, the results show that using both intra- and inter-layer error correction is sensitive to sparsity levels and tends to perform worse at higher sparsity. Specifically, at 20% sparsity, it underperforms compared to intra-layer error correction alone, and at 50% sparsity, it even performs worse than the no-error-correction approach.

To explain why the use of both intra- and inter-layer error correction is sensitive to sparsity levels, we believe this occurs because higher sparsity levels make the pruning task more difficult, leading to greater error accumulation across layers. When both intra- and inter-layer error correction are applied, mitigating the accumulated error from previous layers may dominate the optimization objective in deeper layers, causing the pruning performance of the current layer to suffer.

Mathematically, let $\boldsymbol{W}_k$ and $\boldsymbol{X}_k$ represent the weight matrix and the activation of the $k$-th layer in the original network, respectively. Similarly, let $\boldsymbol{W}_k^*$ and $\boldsymbol{X}_k^*$ denote the pruned weight matrix and the corresponding activation in the pruned network. In a layer-wise pruning scheme with both intra- and inter-layer error correction mechanisms, we minimize the loss for each layer individually:

$$\|\boldsymbol{W}_k^* \boldsymbol{X}_k^* - \boldsymbol{W}_k \boldsymbol{X}_k\|_F^2. \tag{14}$$

$\boldsymbol{X}_k$ depends on the activation from the previous layer:

$$\boldsymbol{X}_k = f_k(\boldsymbol{W}_{k-1} \boldsymbol{X}_{k-1}), \tag{15}$$

where $f_k$ represents some operations (e.g., activation function or normalization). Therefore, we can express the pruned activations recursively as:

$$\boldsymbol{X}_k^* = f_k(\boldsymbol{W}_{k-1}^* \boldsymbol{X}_{k-1}^*). \tag{16}$$

The error at layer $k$ is defined as:

$$\Delta \boldsymbol{X}_k = f_k(\boldsymbol{W}_{k-1}^* \boldsymbol{X}_{k-1}^*) - f_k(\boldsymbol{W}_{k-1} \boldsymbol{X}_{k-1}). \tag{17}$$

Under high sparsity levels, this amplification often results in the accumulated error $\Delta \boldsymbol{X}_k$ becoming dominant at deeper layers. Thus, for large $k$, considering both intra- and inter-layer error correction

mechanisms, we have:

$$\|\boldsymbol{W}_k^*(\boldsymbol{X}_k + \Delta\boldsymbol{X}_k) - \boldsymbol{W}_k\boldsymbol{X}_k\|_F^2 \approx \|\boldsymbol{W}_k^*\Delta\boldsymbol{X}_k - \boldsymbol{W}_k\boldsymbol{X}_k\|_F^2. \tag{18}$$

As a result, the optimization process shifts focus towards correcting this accumulated error rather than pruning the current weight matrix $W_k$.

In other words, minimizing the term in Equation 18 primarily addresses the error correction from previous layers rather than properly pruning the weight matrix $W_k$, which negatively impacts the pruning performance in deeper layers.

