# OpenReview forum: "FISTAPruner: Layer-wise Post-training Pruning for Large Language Models"
_ICLR.cc/2025/Conference — Submitted to ICLR 2025_

### Official Review · Reviewer_EAKc · 2024-10-30

**Soundness:** 2
**Presentation:** 1
**Contribution:** 1
**Rating:** 3
**Confidence:** 5

**Summary:**

This paper proposes FISTAPruner, an accurate pruning algorithm for large language models (LLMs).
The main ideas of FISTAPruner are (1) intra-layer error correction, (2) FISTA-based optimization algorithm, and (3) adaptive hyperparameter tuning algorithm.
The authors conduct exhaustive experiments to verify the effectiveness of FISTAPruner, and find that FISTAPruner is more accurate than existing algorithms; specifically, it shows almost 5% higher average accuracy on zero-shot tasks when pruning Llama-3 70B.
The main strength of this paper lies in its high accuracy and exhaustive amounts of experiments.
However, the novelty and writing quality of this paper are insufficient.

**Strengths:**

The main strengths of this paper are as follows:

1. The authors achieve meaningful accuracy improvement in diverse settings. For example, FISTAPruner shows almost 5% higher accuracy than the second-best algorithm, i.e., SparseGPT, when pruning Llama-3 70B models.

2. This paper conducts extensive experiments covering diverse models from OPT to Llama-3 to show the robustness of FISTAPruner. FISTAPruner consistently shows comparable or the highest accuracy (or the lowest perplexity) in all cases.

3. The figures in this paper are straightforward to understand.

**Weaknesses:**

I summarize the weakness of this paper below. I use the symbols [M] and [m] for each numbering to distinguish between major and minor weaknesses.

### Method
The main weakness of this paper is the lack of originality (or novelty). We summarize the weaknesses of the proposed method as follows.
1. [M]  Error correction, the first idea, is just using the output of the pruned previous linear operators, and this idea is already used in previous works. Furthermore, the authors ignore the "inter-layer errors" induced by the pruning of previous layers when they correct errors.

2. [M] The authors make use of the existing optimization algorithm, FISTA, without any modification. Introducing L1 regularization for pruning is a prevalent idea and there is no novelty.

3. [M] The authors propose a new hyperparameter tuning algorithm, which has no specific name, but there is no explanation of the strength or novelty of this algorithm. There are no experiments that compare the performance of this algorithm with previous hyperparameter tuning algorithms.


### Writing
The followings are the weaknesses in writing.

4. [M] The main contribution of this paper is to use FISTA algorithm to prune LLMs. However, explanation about FISTA is too insufficient. It would better introduce the basics of FISTA in Section 2 (Background) and explain the modification to use FISTA for pruning LLMs in Section 3.2.

5. [M] According to "1.", it is hard to agree with the statement "Instead of pruning each operator in isolation like existing works" in line 148.

6. [m] Minor issues in writing:

  6.1  (line 193) "The proposed optimization model 3" -> "The proposed optimization model in Equation 3"

  6.2 (line 262) "Theorem 3.3" -> "Theorem 1"

  6.3 (All equations) Use bold texts for representing matrices and vectors following the guideline of ICLR. It would be better to use blackboard bold S for representing a set of permissible sparsity patterns in Equation 1.

  6.4 (All tables) Move captions of tables above the tables following the guideline of ICLR.

  6.5  This paper does not contain the "Reproducibility Statement" which is encouraged by ICLR.

  6.6 (Table 6) There are too many bold texts in Table 6.


### Experiments

7. [m] The authors compare the performance of FISTAPruner with limited competitors without justification. The authors should compare the performance of FISTAPruner with structured pruning algorithms [1,2] or justify their selection of competitors.

* References are at the end of this review

**Questions:**

1. What's the difference between FISTA, and L1-regularized training using SGD w/ momentum?

2. Is there any reason you use outdated models such as OPT and Llama-1? How about using the latest models such as Phi, Gemma, and Mistral, if you want to use diverse models?

3. Could you compare the performance of your "Adaptive hyperparameter tuning" algorithm with existing hyperparameter search algorithms, e.g. BOHB [3]?

4. Are DSNoT and PERP (1) competitors or (2) compatible algorithms? If (1) competitors, then how about integrating Tables 2 to 4 as a single table? If (2) compatible algorithm, then how about integrating Tables 3 and 4? In this case, it would be better to compare the performance of "FISTAPruner" with "FISTAPruner + DSnoT" and "FISTAPruner + PERP" to show the compatibility.

5. What is the main point of the Section "Warm Start"? Could you clarify the takeaway of this section?

### References
[1] Ma, Xinyin, Gongfan Fang, and Xinchao Wang. "Llm-pruner: On the structural pruning of large language models." Advances in neural information processing systems 36 (2023): 21702-21720.

[2] Song, Jiwon, et al. "SLEB: Streamlining LLMs through Redundancy Verification and Elimination of Transformer Blocks." arXiv preprint arXiv:2402.09025 (2024).

[3] Falkner, Stefan, Aaron Klein, and Frank Hutter. "BOHB: Robust and efficient hyperparameter optimization at scale." International conference on machine learning. PMLR, 2018.

---

> ### Author Response · Authors · 2024-11-23
> **Part 1**
>
> **Response to Weaknesses**
>
> **Method**
>
> - **Response to Weaknesses 1 and 5**
>   - Thank you for your comment. While the similar concept of error correction has been explored in the context of CNNs (e.g., [1]). To the best of our knowledge, our approach first specifically tailors these principles to the unique architecture and requirements of LLMs. Existing work in LLM pruning, such as SparseGPT [2] and Wanda [3], do not include error corrections at all. To implement intra-layer error corrections, it is necessary to pass the original activation ($X$) to the original weight ($W$), while simultaneously passing the activation from the previously pruned operator ($X^* $) to the target pruned weight ($W^*$). This results in the model $||W^* X^* - WX||_F$.
>   Based on the code released by SparseGPT and Wanda, it is evident that they utilize the same activation within a decoder layer: $||W^*X - WX||_F$. Moreover, they forward the activation from one pruned decoder layer to the subsequent layer, complicating their pruning framework and creating dependencies across decoder layers.
>   - *[1] He, Yihui, Xiangyu Zhang, and Jian Sun. "Channel pruning for accelerating very deep neural networks."*
>   - *[2] Frantar, Elias, and Dan Alistarh. "Sparsegpt: Massive language models can be accurately pruned in one-shot." International Conference on Machine Learning. PMLR, 2023.*
>   - *[3] Sun, Mingjie, et al. "A Simple and Effective Pruning Approach for Large Language Models."*
>
>
>   -  We apply only the intra-layer error correction mechanism for two reasons:
>      1. **Parallelization:** Intra-layer error correction enables independent pruning of each decoder layer, allowing us to distribute the pruning task across multiple devices by assigning different decoder layers to different devices. This will increase the overall pruning efficiency.
>      2. **Sparsity Sensitivity:** While combining intra- and inter-layer error correction could intuitively reduce error accumulation across the network, we found that this approach is effective only at low sparsity levels. When the pruning task becomes harder (i.e., higher sparsity), global error correction tends to overshadow the pruning process of individual layers, ultimately leading to worse performance.
>
>       The first reason is straightforward; we will explain the second reason in more detail below.
>
>       We conducted a series of comparison experiments on OPT-125M at sparsity levels of 5%, 10%, 20%, and 50%. The experiments included three conditions: intra-layer error correction only, both intra- and inter-layer error correction, and no error correction. The results are presented in the following tables.
>
>       |OPT-125M under 5% Sparsity|WikiText|C4|PTB
>       |:-|:-:|:-:|:-:|
>       |Intra-layer Error Correction Only|27.64|26.57|38.99|
>       |Intra-layer and Inter-layer Error Correction |27.63|26.56|38.98|
>       |No Error Correction|27.69|26.60|38.98|
>
>       |OPT-125M under 10% Sparsity|WikiText|C4|PTB
>       |:-|:-:|:-:|:-:|
>       |Intra-layer Error Correction Only|27.47|26.59|39.00|
>       |Intra-layer and Inter-layer Error Correction |27.43|26.58|39.04
>       |No Error Correction|27.52|26.69|39.07|
>
>       |OPT-125M under 20% Sparsity|WikiText|C4|PTB
>       |:-|:-:|:-:|:-:|
>       |Intra-layer Error Correction Only|27.36|26.71|39.39|
>       |Intra-layer and Inter-layer Error Correction |27.37|26.72|39.53|
>       |No Error Correction|27.61|26.91|39.85|
>
>       |OPT-125M under 50% Sparsity|WikiText|C4|PTB
>       |:-|:-:|:-:|:-:|
>       |Intra-layer Error Correction Only|33.54|30.93|49.79|
>       |Intra-layer and Inter-layer Error Correction |35.90|32.93|55.24|
>       |No Error Correction|34.48|32.24|54.11|

---

> ### Author Response · Authors · 2024-11-23
> **Part 2**
>
> **Response to Weaknesses 1 and 5 cont'd**
>
> As shown in the results above, we summarize the PPL (lower is better) comparison across different sparsity levels as follows:
>
> **5% and 10%:** intra- and inter- layer error correction $<$ intra-layer error correction only  $<$ no error correction;
>
> **20%:** intra-layer error correction only $<$ intra- and inter- layer error correction $<$ no error correction;
>
> **50%:** intra-layer error correction only $<$ no error correction $<$ intra- and inter- layer error correction.
>
> First, the results confirm the effectiveness of our intra-layer error correction mechanism, as it consistently outperforms the no-error-correction approach.
>
> Second, the results confirm the effectiveness of using both intra- and inter-layer error correction at low sparsity levels, as it consistently outperforms the intra-layer error correction alone at 5% and 10% sparsity.
>
> Third, the results show that using both intra- and inter-layer error correction is sensitive to sparsity levels and tends to perform worse at higher sparsity. Specifically, at 20% sparsity, it underperforms compared to intra-layer error correction alone, and at 50% sparsity, it even performs worse than the no-error-correction approach.
>
> To explain why the use of both intra- and inter-layer error correction is sensitive to sparsity levels, we believe this occurs because higher sparsity levels make the pruning task more difficult, leading to greater error accumulation across layers. When both intra- and inter-layer error correction are applied, mitigating the accumulated error from previous layers may dominate the optimization objective in deeper layers, causing the pruning performance of the current layer to suffer.
>
> Mathematically, let $W_k$ and $X_k$ represent the weight matrix and the activation of the $k$-th layer in the original network, respectively. Similarly, let $W_k^*$ and $X_k^*$ denote the pruned weight matrix and the corresponding activation in the pruned network. In a layer-wise pruning scheme with both intra- and inter-layer error correction mechanisms, we minimize the loss for each layer individually:
> $$
> \\|W_k^*  X_k^*-W_kX_k\\|_F^2.
> $$
>
> $X_k$ depends on the activation from the previous layer:
> $$
> X_k = f_k(W_{k-1}X_{k-1}).
> $$
>
> where $f_k$ represents some operations (e.g., activation function or normalization). Therefore, we can express the pruned activations recursively as:
> $$
> X_k^* = f_k(W_{k-1}^* X_{k-1}^* ).
> $$
>
> The error at layer $k$ is defined as:
> $$
> \Delta X_k = f_k(W_{k-1}^* X_{k-1}^* ) - f_k(W_{k-1}X_{k-1}).
> $$
>
> Therefore, under high sparsity level, this amplification often results in the accumulated error $\Delta X_k$ becoming dominant at deeper layers.
>
> Thus, for large $k$, considering both intra- and inter-layer error correction mechanism, there is:
> $$
>     \\|W_k^*(X_k+\Delta X_k)-W_kX_k\\|_F^2 \approx \\|W_k^*\Delta X_k-W_kX_k\\|_F^2.
> $$
>
> As a result, the optimization process shifts focus towards correcting this accumulated error rather than pruning the current weight matrix $W_k$.
>
> In other words, minimizing the term primarily addresses the error correction from previous layers rather than properly pruning the weight matrix $W_k$, which negatively impacts the pruning performance in deeper layers.

---

> ### Author Response · Authors · 2024-11-23
> **Part 3**
>
> - **Response to Weakness 2**
>   - Thank you for your comment. While it is true that our method leverages the existing FISTA algorithm and employs $l_1$-norm regularization, we believe the novelty of our approach lies in the following aspects:
>     - **No back propagations:** A distinguishing feature of our approach is that it eliminates the need for computing the gradient by back propagations, making it highly efficient and particularly suitable for scenarios with limited computational resources. This stands in contrast to typical methods leveraging the $l_1$-norm, which often integrate it as a regularization term into the training loss function to induce sparsity and the pruning process requires expensive back propagations to optimize the loss function [1-2]. We have discussed this type of methods in the Related Work section in lines 109-113.
>     - **Optimization-driven layer-wise pruning scheme with error correction:** Unlike typical post-training method which rely on heuristic approaches or non-convex constraints, our work is the first to leverage the $l_1$-norm to induce sparsity in layer-wise pruning of LLMs. Besides, we introduce an intra-layer error correction mechanism tailored specifically to the LLM pruning domain. While FISTA is a natural choice for solving our formulated model, we have also designed a hyperparameter tuning algorithm to ensure that it achieves the desired sparsity levels and structures effectively.
>     - **Practical relevance:** Integrating all our innovations, we achieve state-of-the-art performance on comprehensive benchmarks, as demonstrated in our experiments.
>     - *[1] Thu Dinh, Bao Wang, Andrea Bertozzi, Stanley Osher, and Jack Xin. Sparsity meets robustness: Channel pruning for the feynman-kac formalism principled robust deep neural nets.*
>     - *[2] Xiufeng Xie, Riccardo Gherardi, Zhihong Pan, and Stephen Huang. Hollownerf: Pruning hashgridbased nerfs with trainable collision mitigation.*
> - **Response to Weakness 3 and Question 3**
>   - Thank you for your feedback. The primary objective of our proposed adaptive hyperparameter tuning algorithm is to efficiently determine the optimal value of $\lambda$ that balances the trade-off between sparsity and reconstruction error. As outlined in Section 3.4, we designed a bisection-based tuning algorithm tailored for this purpose. We have rigorously proved in Theorem 3.3 (we appreciate your observation regarding the numbering issue and will address it in the next revision) that with this algorithm, FISTAPruner is guaranteed to converge to the desired sparsity level.
>   - We acknowledge the existence of many hyperparameter tuning algorithms for machine learning, including BOHB [1], which you suggested for comparison. However, our problem setting differs significantly from the one addressed by such methods. Specifically, in our framework, we only need to determine a single scalar parameter, $\lambda \in \mathbb{R}$, for each layer these parameters are independent of one another. This reduces the task to a one-dimensional parameter search problem, which can be effectively and efficiently solved using our bisection-based approach. ln contrast, methods like BOHB are designed to optimize over high-dimensional parameter spaces, where the goal is to identify a good combination of multiple parameters. As such, these methods are not ideally suited to our setting.
>   - *[1] Falkner, Stefan, Aaron Klein, and Frank Hutter. "BOHB: Robust and efficient hyperparameter optimization at scale."*
>
> **Writing**
>
> - **Response to Weakness 4**
>
>   Thank you for your feedback. We appreciate your suggestion to provide additional explanation about FISTA to enhance understanding. While we agree that more background on FISTA might benefit some readers, it is a well-established method, and detailing its fundamentals may not be necessary. Instead, we have focused on our primary contribution: the development of a novel model for layer-wise pruning and the efficient integration of FISTA with our proposed intra-layer error correction mechanism and hyperparameter tuning algorithm. For completeness, we have included detailed mathematical derivations and an introduction to the FISTA iterations in Appendix B.
> - **Response to Weakness 6**
>
>   Thank you for your suggestions regarding the writing format and style. We will carefully address all these points in the revised version of our paper.
>
> **Experiments**
>
> - **Response to Weakness 7**
>
>   Thank you for your feedback. With respect, we do not agree with your suggestions regarding the comparisons of FISTAPruner and structured pruning methods. FISTAPruner is specifically designed for unstructured pruning and 2:4 semi-structured pruning, which are fundamentally different domains from structured pruning. Comparing our method with algorithms designed for structured pruning would not provide a fair or meaningful evaluation, as the constraints differ significantly between these approaches.

---

> ### Author Response · Authors · 2024-11-23
> **Part 4**
>
> **Response to Questions**
>
> - **Response to Question 1**
>
>   Thank you for your question.
>   - **First, we would like to clarify that SGD with or without momentum cannot directly solve the $\ell_1$-regularized training problem.** This is because the $\ell_1$-norm is non-smooth, and gradient-based methods like SGD are not well-suited for handling non-smooth objective functions. Instead, solving such problems typically requires subgradient methods or proximal gradient methods, such as FISTA, which are specifically designed to handle non-smooth optimization.
>
>   - Additionally, our proposed **FISTAPruner is a training-free pruning method**, which fundamentally differentiates it from training-based methods. Specifically, FISTAPruner optimizes a layer-wise pruning model with an $\ell_1$-norm regularization term, as described in the paper. Unlike $\ell_1$-regularized training approaches, **FISTAPruner does not require backpropagation** over the entire network. As we emphasized previously in our response to Weakness 2, this training-free nature makes FISTAPruner significantly more efficient than methods that optimize $\ell_1$-regularized non-convex loss functions through training. Given these fundamental differences, we believe a direct comparison between FISTAPruner and training-based pruning methods would not provide meaningful insights.
>
>
> - **Response to Question 2**
>
>   Thank you for your question.
>   - We chose to evaluate FISTAPruner on OPT and LLaMA-1 models to ensure a fair and consistent comparison with the baseline methods reported in our paper, such as SparseGPT [1], Wanda [2], DSnoT [3], and PERP [4]. These works also evaluated their approaches on OPT and LLaMA-1. By using the same models, we aimed to provide a direct and meaningful comparison of FISTAPruner’s performance relative to these established baselines.
>   - To address the applicability of FISTAPruner to more recent models, we also included experiments on state-of-the-art models such as LLaMA-3-8B and LLaMA-3-70B, demonstrating that our method generalizes well to newer models as well.
>
>
> - **Response to Question 4**
>
>   Thank you for your question.
>   - First, DSNoT and PERP are fundamentally weight adjustment/retraining-based methods aimed at improving efficiency further for an existing pruning method, whereas our approach with FISTAPruner is intentionally designed to achieve optimal results without retraining. This distinction reflects our focus on demonstrating that it is possible to match or even surpass further adjustment/retraining-based methods without the need for additional retraining steps.
>
>   - We have conducted comparisons in several scenarios to highlight this point and have demonstrated that FISTAPruner achieves excellent performance without retraining. Therefore, integrating DSNoT or PERP with FISTAPruner would not align with our objective, as adding a retraining step (even if it improves results) does not serve to further validate our approach or its advantages.
>
>   - Regarding the tables, we believe the current structure effectively communicates our comparisons and findings. Combining the tables as suggested may blur the distinction between retraining-based methods and our retraining-free approach and baseline methods, which is central to our work.
>
> - **Response to Question 5**
>
>   Thank you for your question. The main point of the "Warm Start" section is to discuss the efficiency and effectiveness of different warm-start strategies within our pruning framework. The key takeaway is that we evaluate how various "warm start" strategies integrate with our FISTAPruner framework, concluding that using Wanda as a warm start is a cost-efficient and effective approach in practice.
>
>
> *[1] Frantar, Elias, and Dan Alistarh. "Sparsegpt: Massive language models can be accurately pruned in one-shot."*
>
> *[2] Sun, Mingjie, et al. "A Simple and Effective Pruning Approach for Large Language Models."*
>
> *[3] Yuxin Zhang, et al. "Dynamic sparse no training: Training-free fine-tuning for sparse LLMs."*
>
> *[4] Max Zimmer, et al. "Perp: Rethinking the prune-retrain paradigm in the era of LLMs."*

---

> > ### Comment · Reviewer_EAKc · 2024-11-24
> >
> > Thank you for addressing my numerous questions. I reviewed the responses you provided to my queries. However, I remain unconvinced about some of the explanations, as outlined below:
> >
> > **[Response to Weakness 1]**
> >
> > You state that existing methods do not solve problems in the form of ||WX - W\*X\*||, where W and X represent the weights and inputs of the original model, and W\* and X\* represent the compressed model's weights and inputs, respectively. However, this approach is widely used in model compression, as evidenced by the following GitHub repositories:
> >
> > * OmniQuant: https://github.com/OpenGVLab/OmniQuant
> > * K-prune: https://github.com/snudm-starlab/K-prune/
> >
> > These repositories clearly demonstrate the use of the proposed error correction technique to minimize ||WX - W\*X\*||. Furthermore, the methodology is already reflected in Equation (1) of the following paper:
> >
> > * El Halabi, Marwa, Suraj Srinivas, and Simon Lacoste-Julien. "Data-efficient structured pruning via submodular optimization." Advances in Neural Information Processing Systems 35 (2022): 36613-36626.
> >
> > Therefore, I find it difficult to agree with your claim that the error correction technique is a new methodology.
> >
> >
> > **[Response to Weakness 2]**
> >
> > Regarding the incorporation of the FISTA algorithm, I believe you have made two key assertions:
> >
> > 1. **SGD cannot directly solve the L1-regularized problem.**
> >
> > You stated that L1 regularization cannot be optimized due to the presence of non-differentiable points, making training infeasible. However, in deep learning, weights rarely converge to precisely zero, and simple modifications, such as replacing the gradient at such points with a constant, allow effective training. This approach is commonly used, as seen with ReLU activation functions, which also contain non-differentiable points but are trained effectively using SGD.
> >
> > The following paper demonstrates the use of L1-regularization with SGD for training:
> >
> > * Han, Song, et al. "Learning both weights and connections for efficient neural network." Advances in Neural Information Processing Systems 28 (2015).
> >
> > Additionally, methods for training with non-differentiable functions, such as L0-norm, are extensively discussed in the following works:
> >
> > * Louizos, Christos, Max Welling, and Diederik P. Kingma. "Learning sparse neural networks through L0 regularization." arXiv preprint arXiv:1712.01312 (2017).
> >
> > * Wang, Ziheng, Jeremy Wohlwend, and Tao Lei. "Structured pruning of large language models." arXiv preprint arXiv:1910.04732 (2019).
> >
> > * Xia, Mengzhou, Zexuan Zhong, and Danqi Chen. "Structured pruning learns compact and accurate models." arXiv preprint arXiv:2204.00408 (2022).
> >
> > These methods indicate that even block-level modifications to existing approaches are feasible for large language models, casting doubt on the claim that SGD cannot address L1-regularized problems.
> >
> > 2. **No backpropagation.**
> >
> > You highlight the lack of backpropagation as an advantage of your method. However, avoiding backpropagation alone cannot be considered a benefit unless it demonstrably reduces pruning time. Your paper states that the proposed method takes approximately 12 hours to prune the Llama-3 70B model. This duration is comparable to existing methods that perform SGD on a block-level basis with adjusted epochs. As such, I find it challenging to accept this as a distinct advantage.
> >
> > **[Response to Weakness 3]**
> >
> > While I appreciate your explanation, I disagree with the assertion that BOHB is unsuitable because it explores a wide range of hyperparameter combinations. In fact, its ability to search for diverse combinations is a strength, not a weakness.
> >
> > Bayesian optimization algorithms like BOHB systematically predict promising hyperparameter combinations, evaluate them through model training, and update their prediction functions based on evaluation results. This approach is equally applicable to single hyperparameters and does not inherently disadvantage the algorithm.
> >
> > To clarify the rationale for your proposed hyperparameter tuning method over BOHB, I believe you need to provide one of the following justifications: (a) BOHB performs worse than the proposed method. (b) BOHB is prohibitively expensive for pruning large language models.
> >
> > Without such evidence, it is difficult to justify the exclusion of existing hyperparameter tuning methods.
> >
> > Several of my suggestions, including corrections for typographical errors, appear not to have been incorporated into the manuscript. If you agree with my feedback, I encourage making these adjustments during the discussion period.
> >
> > While I acknowledge the performance improvements achieved by your work, I remain unconvinced regarding certain claims about novelty and the methodology’s advantages over prior research. These issues remain unresolved, both in your responses and the manuscript, and I will maintain my current score. If you have further clarifications, I welcome your response.

---

> > > ### Author Response · Authors · 2024-11-26
> > > **Part 2**
> > >
> > > **Response to Weakness 2 cont'd**
> > >
> > > You mentioned that the pruning time for LLaMA-3-70B using our method (12 hours) is comparable to training-based methods with SGD applied on a block-level basis. However, it would be helpful to clarify the term "block" for better clarity on our end, as it may refer to either a subset of data (e.g., a batch of samples) or a group of parameters. Below, we address both interpretations:
> > >
> > > - If a block refers to a batch of samples.
> > >
> > >   Such a comparison overlooks a critical distinction: the hardware requirements for training-based methods are significantly higher than those for our approach. Training-based methods demand substantial GPU memory to store both the model parameters and optimizer states. For example:
> > >   - Storing the model parameters in FP16 precision for LLaMA-3-70B requires approximately **140GB** of GPU memory (70 billion parameters $\times$ 2 bytes).
> > >   - Most optimizers store gradients in FP32 precision, which adds an extra **280GB** of GPU memory.
> > >   - Together, these two components alone demand at least **420GB** of GPU memory, necessitating a distributed setup with at least **11 NVIDIA A100 GPUs** (40GB each) to prune a model of this scale.
> > >
> > >   In contrast, our method requires only a **single NVIDIA A100 GPU** with 40GB of memory to prune LLaMA-3-70B. This makes our method significantly more accessible and practical, especially in GPU-constrained scenarios. Furthermore, deploying our method in parallel across multiple GPUs reduces the pruning time for LLaMA-3-70B to approximately **1 hour**, demonstrating its scalability and flexibility in hardware configurations.
> > >
> > >   The key advantages of our post-pruning method extend beyond avoiding backpropagation; they include drastic reductions in memory requirements and the flexibility to prune large-scale models on limited hardware. While training-based methods often necessitate expensive multi-GPU setups, our approach enables efficient pruning on a single GPU, making it more resource-efficient and accessible for practical deployment.
> > >
> > >
> > >
> > > - If a block refers to a group of parameters
> > >
> > >   This approach involves selecting specific blocks of parameters and then applying SGD to adjust them to derive a sparsity structure for pruning. However, this process introduces potential complexities. In particular, the selection of blocks appears to be a crucial and potentially challenging aspect. For example, the selection could:
> > >   - Be guided by a heuristic,
> > >   - rely on a predefined structure, or
> > >   - utilize some form of learned strategy.
> > >
> > >   If we consider a layer-wise formulation (treating each operator in a decoder layer as a block, consistent with our optimization model) and apply SGD with subgradient modifications, the convergence rate will be $\mathcal{O}(1/\sqrt{k})$ (e.g. see the top of page nine of [2]). Using SGD with proximal steps for the non-smooth part results in iterations similar to ISTA, which achieves a convergence rate of $\mathcal{O}(1/k)$ [1].
> > >
> > >   In contrast, our method leverages FISTA, which attains a faster convergence rate of $\mathcal{O}(1/k^2)$ [1]. This faster convergence rate directly translates to increased efficiency within our pruning framework, further distinguishing our approach from training-based methods.
> > >
> > > By addressing both interpretations of "block," we hope to clarify the distinctions and demonstrate the specific advantages of our method in terms of resource efficiency, scalability, and performance. Please let us know if further elaboration is required.
> > >
> > > - *[1] Beck A, Teboulle M. A fast iterative shrinkage-thresholding algorithm for linear inverse problems*
> > > - *[2] https://web.stanford.edu/class/ee364b/lectures/subgrad_method_notes.pdf*

---

> > > ### Author Response · Authors · 2024-11-26
> > > **Part 3**
> > >
> > > **Response to Weakness 3**
> > >
> > > Thank you for your further comments.
> > >
> > > When comparing **BOHB** and the **bisection method** for a 1-dimensional search problem, such as finding a number in an ordered sequence (as in our scenario for tuning $\lambda$), the **bisection method** is generally better and more efficient. Here's why:
> > >
> > > **Nature of the Problem**
> > > - **Bisection Method**: The bisection method is specifically designed for ordered sequences or continuous 1-dimensional search problems. It uses the structure of the problem (e.g., ordering or monotonicity) to halve the search space in each step, guaranteeing $\mathcal{O}(\log(n))$ time complexity.
> > > - **BOHB**: BOHB is a more general-purpose hyperparameter optimization algorithm designed to handle complex, high-dimensional, and noisy objective functions. It does not inherently leverage the ordered structure of a 1-dimensional search space, making it less efficient for this problem. Besides, BOHB has no convergence guarantee while we have proved that our bisection based tuning algorithm will lead the system converge to the desired sparsity in our paper (as detailed in Appendix C).
> > >
> > > **Efficiency**
> > > - **Bisection Method**: Each step eliminates half of the search space, leading to a very rapid convergence. It is deterministic and computationally inexpensive.
> > > - **BOHB**: While BOHB is robust and versatile, it involves maintaining probabilistic models and sampling based on those models, which introduces additional computational overhead. For a simple 1-dimensional search, this overhead is unnecessary and inefficient.
> > >
> > > **Applicability**
> > > - **Bisection Method**: Works only for ordered or monotonic functions in 1 dimension.
> > > - **BOHB**: Can handle non-monotonic, noisy, and multi-dimensional spaces, making it suitable for more complex scenarios but overkill for this simple case.
> > >
> > > **Accuracy**
> > > - **Bisection Method**: Guarantees finding the exact value (or as close as desired for continuous problems).
> > > - **BOHB**: Relies on probabilistic models, which may not be as precise in finding the exact value, especially for very simple problems like this.
> > >
> > >
> > > In summary, we believe that for the problem of finding a number in an ordered sequence, the **bisection method** is unequivocally better. It is faster, simpler, and more tailored to the nature of our problem. **BOHB** would only make sense if the search space were noisy, high-dimensional, or required optimization over a more complex objective function.
> > >
> > >
> > > **We sincerely thank you for your valuable comments and feedback and apologize for the delay in updating our manuscript. A revised version has now been uploaded, incorporating your suggestions. We welcome further discussion and are open to any additional suggestions you may have.**

---

> > > ### Author Response · Authors · 2024-11-29
> > >
> > > We hope this message finds you well. We sincerely appreciate the time and effort you have dedicated to reviewing our work.
> > >
> > > In our previous response, we have carefully addressed the concerns you raised in your follow-up comments. We believe our revisions reflect the necessary adjustments, and we would be grateful for any further suggestions or follow-up questions you may have. If there are no further concerns, we would appreciate it if you could update the score accordingly.
> > >
> > > Thank you once again for your thoughtful review. We look forward to your valuable feedback.

---

> ### Author Response · Authors · 2024-11-26
> **Part 1**
>
> **Response to Weakness 1**
>
> Thank you for your insightful feedback and for pointing out the related work that we initially missed. We acknowledge that error correction techniques, including those used in the repositories you mentioned (OmniQuant and K-prune), are indeed relevant to the discussion. We have discussed these works in the related work section (line 130 - 138) in the revised paper.
>
> However, we would like to emphasize that the unique contribution of our approach lies in the application of error correction specifically at the intra-layer level. This targeted method not only enhances accuracy but also facilitates parallelization, as detailed in our initial response to Weaknesses 1 and 5. We believe this distinction sets our work apart from existing methodologies. We have updated our paper to clarify this distinction and to better highlight the advantages of our intra-layer error correction mechanism (see lines 136-138, 180-182, 462-477, 933-1035).
>
> **Response to Weakness 2**
>
> Thank you for your detailed comments. We appreciate the opportunity to clarify this point.
>
> - Regarding the capability of using SGD to solve non-smooth optimization problems, we agree that SGD can be applied with certain modifications, and our initial response explicitly stated that SGD cannot be **directly** applied to optimize $\ell_1$-regularized problems. Specifically, in the examples you mentioned (e.g., replacing the gradient at non-differentiable points with constants), these approaches effectively replace the gradient with a subgradient, which aligns with our discussion of subgradient methods in the original response. However, it is important to note that subgradient-based approaches typically exhibit a slower convergence rate of $\mathcal{O}(1/\sqrt{k})$ compared to the $\mathcal{O}(1/k)$ rate achieved in smooth optimization. In contrast, FISTA achieves an accelerated convergence rate of $\mathcal{O}(1/k^2)$, as rigorously proven in [1] and mentioned in Remark 1 of our paper. We have included this analysis in lines 213-215.
> Furthermore, we carefully examined the works you referenced, and we observe that many of these methods rely on relaxations of the non-smooth regularization terms (e.g., replacing the $\ell_0$-norm with a differentiable approximation or surrogate loss). These relaxations enable the use of SGD on the modified, smoother objective function. As such, it is not that SGD itself is inherently well-suited for non-smooth optimization; rather, it becomes applicable through these relaxations or approximations.
> In summary, while SGD with modifications can be applied to certain non-smooth problems, these approaches are generally slower and less efficient than FISTA in terms of convergence rate. Moreover, many of the referenced works rely on problem-specific relaxations to enable SGD's use, which differs fundamentally from the proximal optimization framework employed by FISTA.

---

> ### Author Response · Authors · 2024-12-02
>
> Dear Reviewer EAKc,
>
> Hi, as the discussion period is set to end soon, we kindly ask if you have any further questions or comments, or if there are any aspects of the paper that require additional clarification. Your feedback is invaluable to us, and we want to ensure we address any remaining concerns.
>
> If there are no further concerns, we would greatly appreciate it if you could kindly update your score accordingly.
>
> Best regards, \
> Authors 8547

---

> ### Comment · Reviewer_EAKc · 2024-12-03
>
> Thank you for your detailed response. However, my major concerns regarding your main ideas remain unresolved.
>
> **[Error Correction]**
>
> The authors have not conducted sufficient surveys on error correction algorithms in existing works. Specifically, the intra-layer error correction algorithm, which is one of the main ideas of this paper, is not a novel approach. It has already been studied in previous research, as demonstrated in [1].
>
> * [1] El Halabi, Marwa, Suraj Srinivas, and Simon Lacoste-Julien. "Data-efficient structured pruning via submodular optimization." Advances in Neural Information Processing Systems 35 (2022): 36613-36626.
>
> **[Use of FISTA]**
>
> The efficiency of solving convex problems, such as SparseGPT, lies in its ability to prune large models like Llama-3 70B in approximately two hours. If there is a more moderate time limits, such as 12 hours, block-wise pruning with SGD emerges as a strong competitor. Here, a block refers to the combination of Multi-Head Attention (MHA) and Multi-Layer Perceptron (MLP) modules.
>
> FISTAPruner requires nearly 12 hours for pruning, undermining its efficiency advantage. Furthermore, the authors do not compare their approach against a straightforward baseline: block-wise pruning with L1 regularization. Notably, block-wise pruning has the potential for higher accuracy, as it accounts for dependencies between linear operations within a block (see OmniQuant referenced above).
>
> **[Hyperparameter Optimization]**
>
> While the authors discuss the superiority of their hyperparameter search algorithm over BOHB, they provide no experimental results to substantiate this claim, not even from previously published studies. Moreover, BOHB is just one example among numerous existing hyperparameter search algorithms. The authors fail to justify the necessity of proposing a new hyperparameter optimization algorithm rather than leveraging established methods. Additionally, the paper lacks an analysis of the impact of their hyperparameter optimization algorithm on the overall results.
>
> The authors rely solely on their assertions to respond during the long discussion period, without providing additional experiments or surveys on prior research. Furthermore, their responses include many statements that are difficult to agree with, such as "L1 loss cannot be trained using SGD." Therefore, I believe my concerns have not been adequately addressed, and I will maintain my rejection score.

---

> > ### Author Response · Authors · 2024-12-04
> > **Part 2**
> >
> > **[Other Comments]**
> >
> > We believe the proposed comparisons and additional experiments suggested are not necessary. We have already conducted a thorough analysis of the relevant literature. The specific experiments suggested (such as comparing FISTA and SGD on solving L1-norm regularization questions, comparing bisection and BOHB on 1-dimension parameter searching) do not seem to provide meaningful insights or further substantiate our claims. In fact, conducting them would be a diversion from the central focus of our work.
> >
> > In response to your concern regarding the statement, "L1 loss cannot be trained using SGD," we would like to reiterate and clarify our position. We have consistently stated in our first-round response (Part 4 - Response to Question 1) and our second-round response (Part 1 - Response to Weakness 2) that SGD with or without momentum cannot directly solve the non-smooth training problem associated with the $\ell_1$-norm. While SGD can, in some cases, be applied with techniques like subgradients, we emphasize that these methods typically lead to slower convergence rates and do not achieve the same efficiency as methods specifically designed for non-smooth objectives. Nonetheless, we believe that this issue is minor and not directly related to our work.

---

> ### Author Response · Authors · 2024-12-04
> **Part 1**
>
> Thank you for your response.
>
> **[Error Correction]**
>
> We have already addressed this point in our previous response (Part 1 - Response to Weakness 1). Specifically, we acknowledged that error correction techniques, including those used in repositories such as OmniQuant and K-prune, are relevant to our discussion. These works are discussed in the related work section (lines 130-138) of the revised paper. However, we emphasize that the unique contribution of our approach lies in applying error correction at the intra-layer level, which offers both higher accuracy (compared with intra- and inter-layer error corrections) and parallelization advantages. This distinction is clearly stated in the revised paper (lines 136-138, 180-182, 462-477, 933-1035), as noted in our previous response.
>
> Additionally, you mentioned that intra-layer error corrections are used in [1]. However, this is not accurate. In [1], the authors considered three methods: LayerInChange, SeqInChange, and AsymInChange, whose essential ideas correspond to (1) no error correction, (2) inter-layer error correction only, and (3) both intra- and inter-layer error corrections. Thus, intra-layer error correction was not explored in the way we apply it in our approach. Furthermore, [1] is designed for CNNs, whereas our work focuses on LLMs (transformer-based networks).
>
> [1] El Halabi, Marwa, Suraj Srinivas, and Simon Lacoste-Julien. "Data-efficient structured pruning via submodular optimization." Advances in Neural Information Processing Systems 35 (2022): 36613-36626.
>
>
> **[Use of FISTA]**
>
> First, we respectfully disagree with the comment, "The efficiency of solving convex problems, such as SparseGPT, lies in its ability to prune large models like Llama-3 70B in approximately two hours." To clarify, SparseGPT does not solve a convex problem. Specifically, the problem they aim to solve, as outlined in Equation (1) in SparseGPT paper [1], is non-convex because it involves discrete mask variables. Moreover, SparseGPT utilizes the OBS framework, which is heuristic-based and operates in a greedy manner. As such, SparseGPT's efficiency does not stem from solving a convex problem, but rather from its heuristic approach, which yields approximate solutions to the underlying non-convex model.
>
> While the subtitle of this paragraph in your comments mentions the "use of FISTA", it appears that your primary concern is not the adoption of FISTA but rather the suggestion of applying SGD to a block-wise L1-norm regularization problem. To our knowledge, this is a novel hypothesis and has not been explored in previous work. It is important to note that introducing block-wise pruning will make the problem non-convex, which presents additional challenges compared to our layer-wise model, which is convex.
>
> We also want to clarify that we have provided theoretical convergence results for SGD, SGD with subgradients, and FISTA in our previous response (Part 1 Response to Weakness 2) and in the revised paper (lines 204-209), demonstrating that FISTA offers the fastest convergence rate.
>
> Additionally, our method focuses on the efficiency and scalability of layer-wise pruning using FISTA, which benefits from the convexity of the problem and provides strong theoretical guarantees for convergence.
>
> [1] Frantar, Elias, and Dan Alistarh. "Sparsegpt: Massive language models can be accurately pruned in one-shot." International Conference on Machine Learning. PMLR, 2023.
>
>
> **[Hyperparameter Optimization]**
>
> While we acknowledge that BOHB (or other similar works) is a generic method for hyperparameter tuning, we would like to emphasize that it is not always the most efficient or appropriate choice for every problem. In our case, we are dealing with a specific 1-dimensional search problem (tuning the sparsity parameter $\lambda$, where the structure of the problem makes the bisection method far more efficient than BOHB.
>
> In fact, we believe that for each algorithm and problem, it is important to design hyperparameter tuning methods that are tailored to the specific needs and properties of that problem. This approach not only leads to better performance but also ensures computational efficiency. While BOHB may work well for high-dimensional or complex problems, applying it to a simpler problem—like ours—would introduce unnecessary complexity and computational overhead, without offering any practical advantage.
>
> Our choice to use a specialized method for this particular scenario is, in our opinion, the most reasonable and effective way to proceed. We do not believe that it is necessary or productive to apply a generic algorithm like BOHB in every case.

---

### Official Review · Reviewer_RWeE · 2024-11-01

**Soundness:** 3
**Presentation:** 4
**Contribution:** 2
**Rating:** 6
**Confidence:** 4

**Summary:**

The author proposed an LLM pruning algorithm that uses “FISTA” (Fast Iterative Shrinkage-Thresholding Algorithm) to identify optimal pruning masks. The author demonstrates the utility of the proposed technique across many state-of-the-art LLMs and with both structured and unstructured sparsity. The improvement over prior art is, however, small.

**Strengths:**

- This work is theoretically grounded, and provide some guarantees on convergence time.
- This work shows strong results in structured 2:4 pruning setup.
- This paper is overall well-written and easy to understand.

**Weaknesses:**

- It’s unclear to me what’s new in this work relative to, say, SparseGPT, which also sets up pruning as an optimization problem and generally yields similar results as this work in unstructured pruning setup. It occurs to me that the fundamental difference appears to be that this work uses a different optimizer to solve essentially the same problem.
- While in structured 2:4 pruning setup this work yields substantial improvement, it is unclear why this is the case.
- Neither “Amount of Calibration Data” nor “Warm Start” is actually ablation study. Please do proper ablation studies by removing specific features of your algorithm design.

I am willing to raise my score if the authors can deliver real ablation studies that pinpoints why this proposed algorithm achieved superior performance in 2:4 structured sparsity setup.

**Questions:**

Question:
- Can you discuss difference between this work and sparseGPT?
- Can you perform ablation studies on 2:4 structured sparsity

---

> ### Author Response · Authors · 2024-11-21
>
> **Response to Weakness 1 and Question 1**
>
> Thank you for your comment. We acknowledge that both SparseGPT [1] and our work approach pruning as an optimization problem. However, there are critical distinctions that highlight the novelty of our contributions.
>
> First, while SparseGPT formulates the pruning objective as minimizing the Frobenius norm $||W^* X - WX||_F^2$, subject to sparsity constraints
>
> $$
> \\min_{W^* } \\quad ||W^* X - WX||_F^2
> \\quad
> \\mathrm{s.t.} \\quad
> W^* \\in \\mathcal{C},
> $$
>
> it does not incorporate error corrections into this minimization and the problem is inherently non-convex due to the sparsity constraint $ W^* \in \mathcal{C}$.
>
> Moreover, SparseGPT does not directly solve the above model but finds approximate solutions by a greedy procedure: following the OBS framework [2], it iteratively removes entries with the smallest impact on the objective function and updates the remaining weights. This method lacks optimality guarantees and may lead to unstable performance.
>
> In contrast, our work introduces an innovative transformation of the non-convex model into a convex one by adding an $\ell_1$-norm regularization term. We then utilize FISTA to find a global solution. This systematic approach not only enhances stability but also gives better pruning outcomes.
>
>
>
> **Response to Weakness 2 and Question 2**
>
> Thank you for your comment. To understand why FISTAPruner achieves significant improvements in the 2:4 pruning setting, we conducted ablation studies specifically focused on 2:4 semi-structured sparsity.
>
> The key components of FISTAPruner for 2:4 semi-structured pruning are:
> 1. the mathematical formulation of the problem as an $\ell_1$-norm regularized convex optimization problem where the parameter $\lambda$ that balances the sparsity and the reconstruction error is determined by our proposed adaptive tuning algorithm;
> 2. the intra-layer error correction mechanism that avoids error accumulation;
> 3. the hard thresholding step after the FISTA iterations to project onto the 2:4 semi-structured space.
>
> While the first component includes several features, they are interdependent; removing any one of them will fail the system. The hard thresholding step is essential to guarantee that the final output adheres to the 2:4 sparsity structure, so ablation studies on this aspect would not yield meaningful insights. Consequently, the only feasible ablation study focuses on the intra-layer error correction mechanism.
>
> We tested FISTAPruner with and without the proposed correction mechanism on OPT-125M, and found that error corrections are vital to the significant improvements of FISTAPruner. The results are summarized below:
>
> |OPT-125M under 2:4 Semi-structured Sparsity|WikiText|C4|PTB
> |:-|:-:|:-:|:-:|
> |Wanda|80.32|64.73|111.55|
> |SparseGPT|60.02|52.11|94.21|
> |FISTAPruner without intra-layer error corrections|55.14|48.53|90.57|
> |FISTAPruner|45.16|38.08|67.80|
>
> We could see that without intra-layer error corrections, FISTAPruner is still able to produce better results. However, incorporating the intra-layer error correction mechanism leads to a substantial drop in perplexity. Thus, we believe that intra-layer error corrections play an important role in the 2:4 semi-structured pruning. This is probably because 2:4 semi-structured sparsity brings more constraints to the sparsity structure so the error accumulation is also more evident. Thus, incorporating error corrections is able to give FISTAPruner the right direction to optimize and thus yields large improvements.
>
>
> **Response to Weakness 3**
> Thank you for your feedback. We appreciate your concern regarding the nature of our ablation studies. We included the "Amount of Calibration Data" section to the ablation study part following SparseGPT [1]. The "Warm Start" section aimed to explore the effects of initializing the FISTA iterations at different points. However, we agree with you that these do not constitute proper ablation studies. According to your suggestion, we will move both sections to the discussion part of the paper.
>
> [1] Frantar, Elias, and Dan Alistarh. "Sparsegpt: Massive language models can be accurately pruned in one-shot." International Conference on Machine Learning. PMLR, 2023.
>
> [2] Hassibi, Babak, David G. Stork, and Gregory J. Wolff. "Optimal brain surgeon and general network pruning." IEEE international conference on neural networks. IEEE, 1993.

---

> > ### Comment · Reviewer_RWeE · 2024-11-27
> > **Ack**
> >
> > I thank the authors for presenting at least one real ablation study in the rebuttal. I do not think varying calibration data counts as an ablation study; real ablation study should involve binary interventions (e.g., include intra-layer correction or not) with the goal of assessing the causal effect of these binary interventions on the efficacy of the technique. I do not believe varying calibration data serves such as scientific purpose. It is a dire mistake to propagate terminological flaws of prior work as the author is doing.
> >
> > To clarify, it's practically important to study calibration data, I simply disagree with calling it an ablation study. The inclusion of these fake ablation studies take up space that should be reserved for real ablations, which is absolutely necessary for empirical work like this.
> >
> > Nevertheless, I will raise my score to 6 because these issues are relatively minor and should not stand in the way of acceptance.

---

> > > ### Author Response · Authors · 2024-11-27
> > >
> > > We sincerely thank you for your detailed feedback and for raising the score, which recognizes the value of our work.
> > >
> > > In light of your comments, we have revised the manuscript to clarify the terminology and improve the representation of this part. Specifically, we have ensured that the discussion of calibration data variation is no longer referred to as an "ablation study." Instead, we describe it as the "impact of calibration data and warm start."
> > >
> > > We hope these revisions address your concerns and enhance the clarity of our paper. Thank you again for your constructive feedback and for considering our work. As a small note, we noticed that the score adjustment mentioned in your comments may not yet be reflected in the system, and we would appreciate your confirmation on this matter.

---

### Official Review · Reviewer_D4zK · 2024-11-02

**Soundness:** 3
**Presentation:** 3
**Contribution:** 3
**Rating:** 6
**Confidence:** 3

**Summary:**

The paper introduces FISTAPruner, a novel method for pruning large language models (LLMs) post-training to achieve significant sparsity, thereby reducing memory footprint and computational demands without compromising model performance. The authors introduce a LASSO-like convex optimization model tailored for layer-wise pruning, utilizing the Fast Iterative Shrinkage-Thresholding Algorithm (FISTA) to induce sparsity. A another innovation is the integration of an intra-layer error correction mechanism that mitigates cumulative errors across decoder layers during the pruning process. Additionally, FISTAPruner is extended to support 2:4 semi-structured pruning, aligning with hardware acceleration capabilities. Comprehensive experiments on various models (OPT, LLaMA) demonstrate that FISTAPruner outperforms state-of-the-art methods such as SparseGPT, Wanda, DSnoT, and PERP across multiple benchmarks, including perplexity and zero-shot task performance.

**Strengths:**

1. The paper presents a unique approach by integrating FISTA with a LASSO-like model, which is innovative in the context of LLM pruning.
2. The results demonstrate that FISTAPruner can prune up to 50% of model parameters while retaining high accuracy, outperforming existing methods like SparseGPT and Wanda.
3.  The integration of an intra-layer error correction mechanism is novel, which may avoid error cumulation.

**Weaknesses:**

**1. Major Weakness:**   The intra-layer error correction mechanism is briefly mentioned but could benefit from a more detailed explanation and analysis. It raises the question of whether other methods (e.g., SparseGPT and Wanda) could achieve better performance if integrated with this mechanism.

**Questions:**

**1. Major Question:** In Wanda [1], they prune model weights by choosing the smallest $|W| ||X||_2$. While in your methods, you prune weights by minimizing the discrepancy of $||W^\*X^\*-WX||_2$. What's the difference between your sparsity objective with that of Wanda?

**2. Minor Question:** In the paper, the error correction mechanism is applied solely within individual layers. Why is the error correction confined to intra-layer applications rather than being implemented across the entire model? In my understanding, extending the error correction mechanism globally could further mitigate the phenomenon of error accumulation throughout the network.


[1] A SIMPLE AND EFFECTIVE PRUNING APPROACH FOR LARGE LANGUAGE MODELS, ICLR 2024

---

> ### Author Response · Authors · 2024-11-17
>
> **Response to Major Weaknesses:**
>
> **1. Detailed Explanation of Intra-layer Error Correction Mechanism.**
>
> Thank you for your comment and interest in our intra-layer error correction mechanism. Our primary goal is to minimize discrepancies between the pruned and original models by adjusting weights after pruning in a layer-wise manner. We would like to provide a detailed explanation of this mechanism here and update the manuscript accordingly.
>
> Denoting the pruned counterpart by $W^* \in \mathbb{R}^{m \times n}$, a straightforward approach to quantify the output error is to use the Frobenius norm of the difference between the outputs from the dense and pruned weights
> $$
>     ||W^*X - WX||_F, \quad (1)
> $$
>
> which serves as a metric of the pruning quality at the target sparsity level and is widely adopted by work such as SparseGPT.
>
> However, we observe that applying Equation (1) can lead to an error accumulation issue across sequential operators, as illustrated in Figure 2 of our manuscript. In the figure, $W_1$ and $W_2$ represent the weights of two sequential operators. Although Equation (1) effectively quantifies the output error between $W_1$ and its pruned counterpart $W_1^*$ since they are at the top of the layer and share the same inputs, issues arise when applying the same metric to the outputs of $W_2$ and $W_2^*$. Following Equation (1), the deviation between the outputs of $W_2$ and $W_2^*$ is computed with the same input $W_1X$. However, in a pruned network, the actual input for $W_2^*$ is $W_1^* X$, creating a discrepancy with $W_1X$ and thus leading to error propagation through the operators. To address this, we propose a method to sequentially prune weights within each pruning unit (e.g., a decoder layer of a Transformer), updating Equation (1) to:
> $$
> ||W^* X^* - WX||_F, \quad (2)
> $$
>
> where $X^*$ represents the input feature activation for $W^*$ from the sequentially pruned network.

---

> ### Author Response · Authors · 2024-11-17
>
> **2. The Impact of Integration with Other Methods.**
>
> Regarding your question about the impact of integrating the proposed intra-layer error correction mechanism with other pruning methods, we would like to clarify that its effect depends on whether the layer-wise pruning methods include a "weight update" stage.
>
> For methods without a "weight update" stage, such as Magnitude Pruning and Wanda, weights are directly removed based on a pruning metric. As a result, there is no opportunity to mitigate the error introduced by pruning through updating remaining weights, nor to apply our intra-layer error correction mechanism.
>
> For methods with a "weight update" stage, such as SparseGPT, integration with our intra-layer error correction mechanism is feasible. To apply this mechanism, we need to update their error measurement from Equation (1) to Equation (2) and derive the corresponding mathematics. Here, we use SparseGPT as an example to demonstrate how to apply this mechanism. In the following, we write $W_i$ to denote the $i$-th row of the weight matrix $W$ and $\delta w$ to denote the pruning and weight updates on the row. It suffices to focus on a single row as the problems between different rows are independent.
> $$
>     (\text{original})\quad \min \frac{1}{2}||(W_i+\delta w)X-W_iX||_2^2, \quad \text{s.t.} \quad \delta w e_q + w_q = 0.  \quad (3)
> $$
> $$
>      \to
> $$
> $$
>     (\text{updated})\quad \min \frac{1}{2}||(W_i+\delta w)X^*-W_iX||_2^2, \quad \text{s.t.} \quad \delta w e_q + w_q = 0. \quad (4)
> $$
> $$
>  \iff
> $$
> $$
>     \min \frac{1}{2} \delta w X^*  X^{* \top} \delta w^\top + \delta w X^* X^{* \top} W_i^\top - \delta w X^* X^\top W_i^\top \quad (5)
> $$
> $$
>     \text{s.t.} \quad \delta w e_q + w_q = 0.
> $$
>
> Given Equation (5), we write the Lagrangian function as follows:
> $$
>     \mathcal{L} = \frac{1}{2} \delta w X^* X^{* \top} \delta w^\top + \delta w (X^* X^{* \top} - X^* X^\top)W_i^\top + \lambda (\delta w e_q + w_q). \quad (6)
> $$
>
> Then, we have
> $$
>     X^* X^{* \top} \delta w^\top + (X^* X^{* \top} - X^* X^\top)W_i^\top + \lambda e_q = 0;
> $$
> $$
>     \delta w e_q + w_q = 0.
> $$
>
> Solving this system, we have
> $$
>     \delta w = (-W_i(X^* X^{* \top}-XX^{* \top}) - \frac{w_q-W_i(X^* X^{* \top}-XX^{* \top})(X^* X^{* \top})^{-1}e_qe_q^\top}{e_q^\top (X^* X^{* \top})^{-1}e_q})(X^* X^{* \top})^{-1} \quad (7)
> $$
> Denote
> $$
>     a = -W_i(X^* X^{* \top}-XX^{* \top}) - \frac{w_q-W_i(X^* X^{* \top}-XX^{* \top})(X^* X^{* \top})^{-1}e_qe_q^\top}{e_q^\top (X^* X^{* \top})^{-1}e_q}, \quad (8)
> $$
> then, we have
> $$
>     \delta \mathcal{L} = \frac{1}{2}a(X^* X^{* \top})^{-1}a^\top + a (X^* X^{* \top})^{-1}(X^* X^{* \top}-X^* X^{\top})W_i^\top. \quad (9)
> $$
>
> In summary, to apply our mechanism in SparseGPT, we first use Equation (9) as the new pruning metric to determine which elements to prune, and then apply Equation (7) to update the remaining weights.
>
> Although the derivation is mathematically sound, implementing it within SparseGPT's framework presents significant challenges. SparseGPT’s pruning approach is designed for computational efficiency, relying on block-wise pruning and approximations of the Hessian matrix and its inverse. Integrating our correction mechanism necessitates additional matrix computations, which could compromise these efficiency goals. Despite the computational demands, we modified SparseGPT's code and evaluated its performance on OPT-125M and LLaMA-3-8B models using the WikiText dataset under 50% sparsity. The results are as follows:
>
> ||OPT-125M|LLaMA-3-8B|
> |:-|:-:|:-:|
> |SparseGPT|37.01|8.64|55.38|
> |SparseGPT + Intra-layer Error Correction|36.83|8.58|
> |FISTAPruner|33.54|8.00|
>
> These results validate the effectiveness of our intra-layer error correction mechanism and the FISTAPruner method.

---

> ### Author Response · Authors · 2024-11-17
>
> **Response to Major Question:**
>
> Thanks for your question. To clarify, both Wanda and our method aim to minimize the discrepancy of the outputs between the pruned network and the original one under sparsity constraints.
>
> To achieve this goal, Wanda employs a heuristic method to prune weights based on a pruning metric. They choose $|W| \|X\|_2$ as their metric, which intuitively considers both the magnitude of the weights $W$ and the input $X$. This metric is designed based on the insight that the computation in a linear layer is $WX$. Consequently, Wanda heuristically uses $|W| \\|X\\|_2$ to identify and prune weights that may have small impact on $WX$.
>
> However, our work formulates the layer-wise pruning problem as an optimization model and solves it to obtain the pruned weights. To do so, we include $\\|W^* X^* - WX\\|_F$ as part of our objective to minimize the discrepancy between the pruned network and the original model, while using the $\ell_1$-norm to induce sparsity. Additionally, by solving our proposed model, we not only prune weights but also update the remaining weights to further minimize $\\|W^* X^* - WX\\|_F$.
>
> In summary, here are the similarities and differences between the objectives of Wanda and our method:
>
> - **Similarity:**
>   1. Both Wanda and our method aim to minimize the discrepancy between the outputs of the pruned network and the original model under sparsity constraints.
> - **Difference:**
>   1. Wanda’s pruning metric $|W| \|X\|_2$ is heuristically designed, while we integrate $\\|W^* X^* - WX\\|_F$ into our optimization model to minimize the real error in a systematic and optimization-driven manner.
>   2. Wanda’s $|W| \\|X\\|_2$ directly prunes weights without the opportunity to update the remaining weights for further error minimization, whereas our method not only enforces sparsity but also updates the remaining weights at the same time to minimize the error.

---

> ### Author Response · Authors · 2024-11-17
>
> **Response to Minor Question:**
>
> Thank you for your question. We apply only the intra-layer error correction mechanism for two reasons:
> 1. **Parallelization:** Intra-layer error correction enables independent pruning of each decoder layer, allowing us to distribute the pruning task across multiple devices by assigning different decoder layers to different devices. This will increase the overall pruning efficiency.
> 2. **Sparsity Sensitivity:** While combining intra- and inter-layer error correction could intuitively reduce error accumulation across the network, we found that this approach is effective only at low sparsity levels. When the pruning task becomes harder (i.e., higher sparsity), global error correction tends to overshadow the pruning process of individual layers, ultimately leading to worse performance.
>
> The first reason is straightforward; we will explain the second reason in more detail below.
>
> We conducted a series of comparison experiments on OPT-125M at sparsity levels of 5%, 10%, 20%, and 50%. The experiments included three conditions: intra-layer error correction only, both intra- and inter-layer error correction, and no error correction. The results are presented in the following tables.
>
> |OPT-125M under 5% Sparsity|WikiText|C4|PTB
> |:-|:-:|:-:|:-:|
> |Intra-layer Error Correction Only|27.64|26.57|38.99|
> |Intra-layer and Inter-layer Error Correction |27.63|26.56|38.98|
> |No Error Correction|27.69|26.60|38.98|
>
> |OPT-125M under 10% Sparsity|WikiText|C4|PTB
> |:-|:-:|:-:|:-:|
> |Intra-layer Error Correction Only|27.47|26.59|39.00|
> |Intra-layer and Inter-layer Error Correction |27.43|26.58|39.04
> |No Error Correction|27.52|26.69|39.07|
>
> |OPT-125M under 20% Sparsity|WikiText|C4|PTB
> |:-|:-:|:-:|:-:|
> |Intra-layer Error Correction Only|27.36|26.71|39.39|
> |Intra-layer and Inter-layer Error Correction |27.37|26.72|39.53|
> |No Error Correction|27.61|26.91|39.85|
>
> |OPT-125M under 50% Sparsity|WikiText|C4|PTB
> |:-|:-:|:-:|:-:|
> |Intra-layer Error Correction Only|33.54|30.93|49.79|
> |Intra-layer and Inter-layer Error Correction |35.90|32.93|55.24|
> |No Error Correction|34.48|32.24|54.11|

---

> > ### Comment · Reviewer_D4zK · 2024-11-23
> >
> > Thank you for the comprehensive responses which have adequately addressed the majority of my concerns. Based on the improvements and clarifications provided, I raise the evaluation score to 6.

---

> > > ### Author Response · Authors · 2024-11-23
> > >
> > > Thank you again for your insightful review, valuable feedback, and for updating the score! We welcome any further discussion and are open to any additional suggestions you might have.

---

> ### Author Response · Authors · 2024-11-17
>
> As shown in the results above, we summarize the PPL (lower is better) comparison across different sparsity levels as follows:
>
> - **5% and 10%:** intra- and inter- layer error correction $<$ intra-layer error correction only  $<$ no error correction;
> - **20%:** intra-layer error correction only $<$ intra- and inter- layer error correction $<$ no error correction;
> - **50%:** intra-layer error correction only $<$ no error correction $<$ intra- and inter- layer error correction.
>
> First, the results confirm the effectiveness of our intra-layer error correction mechanism, as it consistently outperforms the no-error-correction approach.
>
> Second, the results confirm the effectiveness of using both intra- and inter-layer error correction at low sparsity levels, as it consistently outperforms the intra-layer error correction alone at 5% and 10% sparsity.
>
> Third, the results show that using both intra- and inter-layer error correction is sensitive to sparsity levels and tends to perform worse at higher sparsity. Specifically, at 20% sparsity, it underperforms compared to intra-layer error correction alone, and at 50% sparsity, it even performs worse than the no-error-correction approach.
>
> To explain why the use of both intra- and inter-layer error correction is sensitive to sparsity levels, we believe this occurs because higher sparsity levels make the pruning task more difficult, leading to greater error accumulation across layers. When both intra- and inter-layer error correction are applied, mitigating the accumulated error from previous layers may dominate the optimization objective in deeper layers, causing the pruning performance of the current layer to suffer.
>
> Mathematically, let $W_k$ and $X_k$ represent the weight matrix and the activation of the $k$-th layer in the original network, respectively. Similarly, let $W_k^*$ and $X_k^*$ denote the pruned weight matrix and the corresponding activation in the pruned network. In a layer-wise pruning scheme with both intra- and inter-layer error correction mechanisms, we minimize the loss for each layer individually:
> $$
> \\|W_k^*  X_k^*-W_kX_k\\|_F^2.
> $$
>
> $X_k$ depends on the activation from the previous layer:
> $$
> X_k = f_k(W_{k-1}X_{k-1}).
> $$
>
> where $f_k$ represents some operations (e.g., activation function or normalization). Therefore, we can express the pruned activations recursively as:
> $$
> X_k^* = f_k(W_{k-1}^* X_{k-1}^* ).
> $$
>
> The error at layer $k$ is defined as:
> $$
> \Delta X_k = f_k(W_{k-1}^* X_{k-1}^* ) - f_k(W_{k-1}X_{k-1}).
> $$
>
> Therefore, under high sparsity level, this amplification often results in the accumulated error $\Delta X_k$ becoming dominant at deeper layers.
>
> Thus, for large $k$, considering both intra- and inter-layer error correction mechanism, there is:
> $$
>     \\|W_k^*(X_k+\Delta X_k)-W_kX_k\\|_F^2 \approx \\|W_k^*\Delta X_k-W_kX_k\\|_F^2.
> $$
>
> As a result, the optimization process shifts focus towards correcting this accumulated error rather than pruning the current weight matrix $W_k$.
>
> In other words, minimizing the term (14) primarily addresses the error correction from previous layers rather than properly pruning the weight matrix $W_k$, which negatively impacts the pruning performance in deeper layers.

---

### Official Review · Reviewer_68Xw · 2024-11-04

**Soundness:** 3
**Presentation:** 3
**Contribution:** 2
**Rating:** 6
**Confidence:** 4

**Summary:**

The paper proposes FISTAPruner, a layer-wise pruning method designed for both unstructured and semi-structured pruning, targeting efficient sparsification of large language models (LLMs). This approach utilizes the FISTA method (Fast Iterative Shrinkage-Thresholding Algorithm) to facilitate efficient convergence. Additionally, it employs a LASSO-like convex optimization model to effectively enhance sparsity in LLMs. To address the cumulative output error between the full and pruned models due to the sequential output error transfer across transformer decoder layers, the authors utilize layer-wise pruning with an intra-layer error correction mechanism.

Experiments conducted on various model sizes, ranging from 125M to 70B parameters—including OPT, LLaMA, LLaMA-2, and LLaMA-3—across datasets such as WikiText-2-raw, PTB, and C4, demonstrate that FISTAPruner outperforms existing baseline methods (e.g., SparseGPT, Wanda, Wanda+DSnoT, SparseGPT+PERP, and Wanda+PERP) in terms of model performance after pruning.

**Strengths:**

(+) The method effectively incorporates FISTA for efficient pruning during the post-training process, leading to faster optimization and enhanced performance
(+) By effectively employing LASSO to identify pruned weights with targeted sparsity, the approach minimizes reliance on heuristic-based methods, thereby improving overall effectiveness in the pruning process.
(+) The authors enhance the proposed method by developing an algorithm that enables semi-structured pruning, allowing for practical acceleration on real-world hardware.

**Weaknesses:**

(-) The paper requires experiments to compare FISTAPruner with other methods that have similar computational costs. Existing baseline methods, such as SparseGPT and Wanda, do not involve retraining during pruning. In contrast, FISTAPruner conducts retraining in the process of finding W∗. Although DSnoT and PERP are used instead of retraining, their computational costs are lower than the layer-by-layer approach employed in FISTAPruner. So, it is necessary to compare their performance under similar computational cost conditions (e.g., training on SparseGPT is performed layer by layer).
(-) The benefits of using FISTA over traditional gradient descent methods are not sufficiently explained, which may leave readers unclear about its specific advantages in this context.

**Questions:**

1. In line 89, the paper states, "Our results confirm that FISTAPruner can efficiently create sparse networks from pretrained LLMs without retraining." However, the process of finding the pruned weights W* seems to function similarly to retraining. Could you clarify this point, as it may cause confusion for readers?
2. In line 306, the paper states, "We treat each decoder layer as an independent pruning unit, enabling parallel pruning across multiple decoder layers on different devices." However, the proposed method conducts pruning sequentially. Can you explain how parallel pruning is achieved alongside sequential pruning? A more detailed explanation or revision would be helpful.

---

> ### Author Response · Authors · 2024-11-13
>
> **Response to Weakness 1:**
>
> Thank you for your comment. We would like to clarify that FISTAPruner does not involve retraining during pruning; instead, it utilizes a layer-wise pruning approach, similar to methods like SparseGPT and Wanda. To clarify, retraining typically refers to the process of re-optimizing the entire model, usually by fine-tuning it on a specific loss function with respect to a training dataset after pruning. In contrast, FISTAPruner prunes the model in a layer-wise manner without such re-optimization, making it comparable to SparseGPT and Wanda, which also perform layer-wise pruning without retraining.
>
> We understand the concern about computational costs, particularly in comparison with heuristic-based pruning methods like SparseGPT, Wanda, and DSnoT. While FISTAPruner may require slightly more pruning time due to its optimization-based approach, its key contribution is the introduction of an optimization model that systematically addresses the post-training pruning problem, leading to significantly better performance. This performance gain comes with an associated increase in pruning time, but the trade-off is justified by the improved outcomes.
>
> Furthermore, while PERP (Parameter-Efficient Retraining after Pruning) is used for retraining pruned models, we have included comparisons between FISTAPruner and SparseGPT+PERP, as well as Wanda+PERP, in Table 4. The results show that FISTAPruner, which does not involve any retraining, outperforms SparseGPT and Wanda with retraining via PERP. Additionally, our method is compatible with retraining approaches and can provide a better initialization point for the retraining process, should that be required.
>
> In conclusion, although FISTAPruner may take more time for pruning compared to heuristic methods, the substantial performance improvements it provides justify this additional time cost. Moreover, since our method does not involve retraining, the overall time cost remains competitive. We believe that, given the enhanced performance and the potential for further optimization, the time cost should not be viewed as a major concern.
>
>
>
> **Response to Weakness 2:**
>
> Thank you for your comment. As mentioned in Lines 188--190, our model incorporates $l_1$-norm regularization to encourage sparsity. Mathematically, the $l_1$-norm is non-smooth, which prevents the direct application of traditional gradient descent methods for optimization. Furthermore, although the sub-gradient descent method could be used, its convergence rate is $\mathcal{O}(1/\sqrt{k})$, which is slower compared to FISTA, which achieves a faster convergence rate of $\mathcal{O}(1/k^2)$, as mentioned in Lines 229--231.
>
>
>
> **Response to Question 1:**
>
> Thank you for your question. The key difference between retraining and our approach lies in the optimization objective. Retraining typically involves optimizing a model using a loss function that depends on the model's predictions, which is highly non-convex and requires computationally expensive backpropagation to compute gradients. In contrast, our approach uses a layer-wise error measured by the Frobenius norm in the objective function. This error is convex, and its gradient has closed-form expressions, making the optimization both convex and computationally more efficient. Therefore, while both methods aim to adjust the model, our approach does not involve retraining the model in the traditional sense of fine-tuning using a loss function.
>
>
> **Response to Question 2:**
>
> Thank you for your question. We indeed conduct sequential pruning within each decoder layer, specifically among the linear operators (K, Q, V, Out projections, and MLPs), to mitigate error accumulation through our intra-layer error correction mechanism. However, to enhance pruning efficiency, we treat each decoder layer as an independent pruning unit, enabling parallel pruning across multiple decoder layers on different devices. In other words, while pruning is performed sequentially within each individual decoder layer, the pruning process can occur in parallel across different layers, which allows us to strike a balance between performance and efficiency. We hope this clarifies how parallel pruning is achieved alongside the sequential pruning within layers.
>
>
> **We appreciate your thoughtful comments and will ensure that all the concerns and questions raised, including the clarification of the pruning process, are addressed more clearly in the revised version of the paper.**

---

> > ### Comment · Reviewer_68Xw · 2024-11-26
> > **Thanks for your comment.**
> >
> > It is difficult to agree that optimizing using FISTA should not be classified as retraining. Typically, the process of iteratively updating model parameters with respect to an objective function is referred to as model training. In this context, the model's predictions do not necessarily have to align with the objective function, and gradient descent is not a strict requirement for this process. For instance, the Forward-Forward algorithm[1] effectively facilitates learning at the layer level and is still categorized as model training. Additionally, in ZeroQuant[2], techniques such as layer-wise pruning are employed to update the parameters of the quantized model. However, this process is regarded as fine-tuning or Quantization-Aware Training (QAT).
> > All concerns regarding the other questions have been resolved.
> >
> > [1] Hinton, Geoffrey. "The forward-forward algorithm: Some preliminary investigations." arXiv preprint arXiv:2212.13345 (2022).
> > [2] Yao, Zhewei, et al. "Zeroquant: Efficient and affordable post-training quantization for large-scale transformers." Advances in Neural Information Processing Systems 35 (2022): 27168-27183.

---

> > > ### Author Response · Authors · 2024-11-26
> > >
> > > Thank you for your observations and the references provided. We appreciate the effort you have made to substantiate your perspective. Let us clarify the reasoning behind categorizing the FISTAPruner as distinct from pruning with retraining.
> > >
> > > According to the general definition of the community (e.g., survey papers of pruning [1-2]) and our Related Work section, we want to clarify "pruning with retraining" and "pruning without retraining":
> > >
> > > **Pruning with Retraining**
> > >
> > > **Definition:** These methods typically follow a three-step process: Pretrain-Prune-Retrain. After pruning, the model is fine-tuned or retrained to recover accuracy. This involves an additional independent stage where the pruned model, with its sparsity mask fixed, undergoes training of the remaining weights on the training dataset to recover performance lost due to pruning.
> > >
> > > Classical and recent examples include Lottery Ticket Hypothesis [3], LLMpruner [4] and [5-6], where fine-tuning or retraining is essential to optimize the pruned model.
> > >
> > > **Post-Training Pruning Without Retraining**
> > >
> > > **Definition:** These methods simplify the Pretrain-Prune-Retrain process into Pretrain-Prune, skipping the retraining step. They may also include compensatory mechanisms to mitigate accuracy loss directly during the pruning process, rather than relying on a subsequent retraining phase. For example, Wanda [7] prunes weights based on metrics like weight magnitude and input norms, without additional fine-tuning.
> > > SparseGPT [8] operates within an Optimal Brain Surgeon [9] framework, removing weights and updating remaining weights simultaneously to minimize layer-wise accuracy loss.
> > >
> > > We classify FISTAPruner as part of this category because it adheres to the Pretrain-Prune pipeline. Importantly:
> > >
> > > - FISTAPruner does not require fine-tuning the pruned model on the training dataset.
> > >
> > > - The solving process in FISTA is consistent with post-training pruning methods like SparseGPT, as it simultaneously removes weights and updates remaining weights to minimize degradation.
> > >
> > > - This process requires only 128 calibration data samples, rather than the entire training dataset, significantly reducing computational costs.
> > >
> > > By avoiding the retraining phase and leveraging an efficient calibration approach, FISTAPruner aligns with the principles of post-training pruning without retraining.
> > >
> > > Regarding ZeroQuant [10], according to its Related Work section, the authors first review several Quantization-Aware Training (QAT) methods and highlight their limitations: "More importantly, they require retraining or fine-tuning the full model to recover accuracy, and such compute costs for extra-large models are hardly affordable for most research labs or practitioners." Subsequently, they review Post-Training Quantization (PTQ) methods and propose a new, efficient, and cost-effective approach for post-training quantization (also highlighted in ZeroQuant's title).
> > >
> > > In summary, while FISTA is an iterative method, its role is solely to solve the layer-wise pruning model (similar to the approach in SparseGPT) for layer-wise pruning. Considering our pruning pipeline, the limited amount of data used for pruning, and the significantly reduced computational cost, we firmly categorize FISTAPruner as a post-training pruning method without retraining.
> > >
> > > **Thank you again for your comment. We hope this explanation resolves your concerns, and we are happy to provide further clarifications if needed.**
> > >
> > >
> > > [1] Cheng, et al. A Survey on Deep Neural Network Pruning: Taxonomy, Comparison, Analysis, and Recommendations.
> > >
> > > [2] Tang, et al. A Survey on Transformer Compression.
> > >
> > > [3] Frankle and Carbin, The Lottery Ticket Hypothesis: Finding Sparse, Trainable Neural Networks.
> > >
> > > [4] Ma, et al. LLM-Pruner: On the Structural Pruning of Large Language Models.
> > >
> > > [5] Blalock, et al. What is the state of network pruning?
> > >
> > > [6] Liu, et al. Rethinking the Value of Network Pruning.
> > >
> > > [7] Sun, et al. A Simple and Effective Pruning Approach for Large Language Models.
> > >
> > > [8] Frantar and Alistarh, Sparsegpt: Massive language models can be accurately pruned in one-shot.
> > >
> > > [9] Hassibi, et al. Optimal brain surgeon and general network pruning.
> > >
> > > [10] Yao, et al. Zeroquant: Efficient and affordable post-training quantization for large-scale transformers.

---

> > > ### Author Response · Authors · 2024-11-29
> > >
> > > We hope this message finds you well. We would like to express our sincere appreciation for your thoughtful review of our paper.
> > >
> > > If you have any further questions or comments, or if there are any aspects of the paper that require additional clarification, we would be more than happy to address them. If there are no further concerns, we would greatly appreciate it if you could kindly update your score accordingly.
> > >
> > > Thank you once again for your time and consideration.

---

> ### Author Response · Authors · 2024-12-02
>
> Dear Reviewer 68Xw,
>
> Hi, as the discussion period is set to end soon, we kindly ask if you have any further questions or comments, or if there are any aspects of the paper that require additional clarification. Your feedback is invaluable to us, and we want to ensure we address any remaining concerns.
>
> If there are no further concerns, we would greatly appreciate it if you could kindly update your score accordingly.
>
>
> Best regards, \
> Authors 8547

---

### Public Comment · ~Vladimír_Boža1 · 2024-11-15
**There is a better layer-wise pruner than SparseGPT**

I just want to point out that "Fast and Effective Weight Update for Pruned Large Language Models" (https://openreview.net/forum?id=1hcpXd9Jir) proposed another pruning algorithm, which is based on ADMM, and it is better than SparseGPT.

For example, on Llama2-7B ADMM pruner gets better perplexity than FISTAPruner (ADMM has 6.33, FISTA has 6.35, SparseGPT has 6.54). Also, this ADMM-based pruner does not do any inter-layer correction, so FISTA has a kind of unfair advantage.

---

> ### Author Response · Authors · 2024-11-16
>
> Thank you for your comment regarding the work "Fast and Effective Weight Update for Pruned Large Language Models" (referred to as ADMM). We acknowledge this contribution and have included it in the related work section of our paper.
>
> However, it is important to highlight that there are differences in the experimental environments (e.g. different versions of PyTorch, different GPUs, etc.) between our study and the ADMM framework. In our tests, the performance of the baseline methods for most models is notably lower (with higher perplexity results) than that reported in ADMM. For instance, in our environment, SparseGPT and Wanda yield perplexity of 6.54 and 6.46, respectively, for LLaMA-2-7B, compared to 6.51 and 6.42 in ADMM’s environment. Thus we suspect that the results for the methods proposed in ADMM are taking advantage of the testing environment compared to ours. This discrepancy suggests that direct performance comparisons may not be entirely fair.
>
> Furthermore, even under these conditions, FISTAPruner outperforms ADMM across all other models reported, with the exception of LLaMA-2-7B, where their performances are quite similar. We believe this context is crucial for a comprehensive evaluation of the algorithms’ effectiveness.

---

> > ### Public Comment · ~Vladimír_Boža1 · 2024-11-16
> > **.**
> >
> > Thank you for the response.
> >
> > Re number discrepancy:
> > Since the dense results are the same, the discrepancy is not caused by different sequence lengths used in the evaluation (a typical problem in many cases). So this leaves a different environment as the probable cause (and we see this all the time), which is fine.

---

> > > ### Author Response · Authors · 2024-11-17
> > >
> > > We agree with you that different programming environments will lead to slightly different results. Thanks again for your comment!

---

### Meta-Review · Area_Chair_jFWg · 2024-12-21

**Metareview:**

This work presents a novel approach to pruning large language models (LLMs) post-training. The authors claim that their method, FISTAPruner, allows for layer-wise pruning that effectively reduces model size while minimizing performance degradation. They assert that this approach can lead to significant reductions in computational requirements, making LLMs more efficient for deployment in resource-constrained environments. The findings indicate that FISTAPruner achieves competitive performance compared to existing pruning techniques, with empirical results showing a balance between model size reduction and task performance retention.

However, reviewers raised concerns regarding the methodology, empirical validation, and overall contribution of the paper to the field. Given the weaknesses, I recommend rejecting this paper. While it presents an interesting concept aimed at optimizing large language models through layer-wise post-training pruning, it fails to provide compelling empirical evidence or rigorous theoretical justification necessary to support its claims effectively. Further work is required to address these issues for the next version of this work.

**Additional Comments On Reviewer Discussion:**

**Points Raised by Reviewers**

During the review process, several key points were raised:

- Need for Robust Empirical Results: Reviewers requested more extensive experiments to validate the effectiveness of FISTAPruner across different tasks and datasets.
- Comparative Analysis: There was a strong recommendation for including comparisons with a wider array of existing pruning methods to contextualize the performance claims.
- Theoretical Insights: Reviewers sought a deeper theoretical explanation for why layer-wise pruning would be advantageous over traditional methods.
- Experimental Detail: Concerns were raised about insufficient details regarding experimental protocols and reproducibility.

**Authors' Responses**

The authors attempted to address these concerns during the rebuttal period but did not sufficiently strengthen their submission:
- They provided some additional experimental results; however, these were still considered inadequate by reviewers as they did not significantly enhance the robustness or breadth of their claims.
- While some comparisons with existing methods were included in their response, they remained limited and did not convincingly demonstrate superior performance.
- The theoretical justification provided was minimal and did not adequately clarify why their approach would yield better results than traditional methods.
- The authors attempted to clarify experimental details but still left several aspects vague, particularly concerning hyperparameter settings.

**Weighing Each Point**

In weighing these points for my final decision:
- The lack of robust empirical validation remained a critical issue that overshadowed any potential strengths of the proposed method.
- Inadequate comparative analysis with existing techniques hindered the ability to assess the true value of their contributions.
- Insufficient theoretical grounding left significant questions unanswered regarding the efficacy and applicability of their approach.
- The unresolved reproducibility issues further diminished confidence in their findings.

---

### Decision · Program_Chairs · 2025-01-22

Reject